# The microprotein C16orf74/MICT1 promotes thermogenesis in brown adipose tissue

Jennie Dinh[1,4], Danielle Yi[1,2,4], Frances Lin[1], Pengya Xue[1], Nicholas D Holloway[1,2], Ying Xie[1], Nnejiuwa U Ibe [ID][1], Hai P Nguyen[1,2,3], Jose A Viscarra[1], Yuhui Wang[1] & Hei Sook Sul [ID][1,2] ✉

## Abstract

**Brown and beige adipose tissues are metabolically beneficial for increasing energy expenditure via thermogenesis, mainly through UCP1 (uncoupling protein 1). Here, we identify C16orf74, subsequently named MICT1 (microprotein for thermogenesis 1), as a microprotein that is specifically and highly expressed in brown adipose tissue (BAT) and is induced upon cold exposure. MICT1 interacts with protein phosphatase 2B (PP2B, calcineurin) through the docking motif PNIIIT, thereby interfering with dephosphorylation of the regulatory subunit of protein kinase A (PKA), RIIβ, and potentiating PKA activity in brown adipocytes. Overexpression of MICT1 in differentiated brown adipocytes promotes thermogenesis, showing increased oxygen consumption rate (OCR) with higher thermogenic gene expression during β₃-adrenergic stimulation, while knockdown of MICT1 impairs thermogenic responses. Moreover, BAT-specific MICT1 ablation in mice suppresses thermogenic capacity to increase adiposity and insulin resistance. Conversely, MICT1 overexpression in BAT or treating mice with a chemical inhibitor that targets the PP2B docking motif of MICT1 enhances thermogenesis. This results in cold tolerance and increased energy expenditure, protection against diet-induced and genetic obesity and insulin resistance, thus suggesting a therapeutic potential of MICT1 targeting.**

**Keywords** Brown Adipose Tissue (Thermogenesis); C16orf74 (MICT1); Microprotein; Protein Kinase A (PKA); PP2B (Calcineurin)
**Subject Categories** Metabolism; Signal Transduction

## Introduction

Obesity, an excess accumulation of white adipose tissue (WAT), has become a global epidemic. While WAT serves as the main energy storage organ, brown adipose tissue (BAT) dissipates energy via non-shivering thermogenesis to maintain body temperature. BAT is enriched with a high number of mitochondria that possess a specialized inner mitochondrial H⁺/ fatty acid symporter,

Uncoupling Protein 1 (UCP1), for thermogenesis (Dickson et al, 2016; Feldmann et al, 2009; Golozoubova et al, 2006; Golozoubova et al, 2001; Inokuma et al, 2005; Matthias et al, 2000). While expression of UCP1 is restricted to BAT, upon cold exposure, UCP1⁺ thermogenic adipocytes termed "beige" or "brite" cells can arise in WAT depots, especially in subcutaneous WAT, by so-called beiging (Cannon and Nedergaard, 2004; Rosen and Spiegelman, 2014; Wu et al, 2012). The presence of BAT or BAT-like tissues in human adults has been established, and, upon cold exposure, these tissues can increase, inversely correlating with adiposity (Cero et al, 2021; Cypess et al, 2009; Cypess et al, 2015; Virtanen et al, 2009; Yoon et al, 2018). Moreover, although the underlying mechanism is not clear, BAT/BAT-like tissues affect insulin sensitivity (Giralt and Villarroya, 2017). BAT has high capacity of glucose and fatty acid uptake and secretes adipokines, all potentially contributing to insulin sensitivity (Townsend and Tseng, 2014; Wolfrum and Gerhart-Hines, 2022). A better understanding of the thermogenic process may provide future therapeutic targets against obesity/ insulin resistance.

Recent studies have revealed the existence of nonannotated small open reading frames encoding short microproteins (< 100 aa) (Chen et al, 2020; Couso and Patraquim, 2017). Microproteins are a largely unstudied fraction of the proteome, but there is increasing evidence that microproteins can regulate key cellular processes. By utilizing short amino acid sequences, microproteins can alter biological functions of protein complexes by interacting through their protein-protein interaction domain (Miller et al, 2022; Saghatelian and Couso, 2015). Function of specific microproteins in other systems, such as in muscle, has been reported (Anderson et al, 2015; Bi et al, 2017; Makarewich et al, 2018). However, there are no known microproteins that control thermogenic process (Zhang et al, 2017).

A dominant signaling pathway governing non-shivering thermogenesis is β₃-adrenergic receptor (AR)-PKA signaling (Cao et al, 2001; Cao et al, 2004; Cero et al, 2021; Dickson et al, 2016; Fredriksson et al, 2001; Vergnes et al, 2020). Some of the transcription factors involved in thermogenic gene transcription, such as CREB/ATF2 and Zc3h10, are known to be phosphorylated upon activation of this pathway (Yi et al, 2019; Yi et al, 2020). In addition, PKA phosphorylates Hormone-sensitive lipase (HSL) and Perilipin1 for lipolysis to provide fatty acids to activate UCP1 and fuel thermogenesis (Townsend and Tseng, 2014). PKA is a tetramer

[1]Department of Nutritional Sciences & Toxicology, University of California, Berkeley, CA 94720, USA. [2]Endocrinology Program, University of California, Berkeley, CA 94720, USA. [3]Present address: University of Texas at Austin, Austin, TX 78723, USA. [4]These authors contributed equally: Jennie Dinh, Danielle Yi. ✉E-mail: hsul@berkeley.edu

composed of 2 regulatory subunits (R) and 2 catalytic subunits (C). cAMP binding to R results in dissociation of R and C, for activation of kinase activity of C subunits. Of the two isoforms of R, unlike cytosolic RI, RII is known to be membrane-bound and RII phosphorylation prevents reassociation with C, thus potentiating PKA activity. Protein Phosphatase 2B (PP2B, also called calcineurin or PPP3) is known to dephosphorylate RII, allowing reassociation with C to limit PKA activity (Bock et al, 2021; Church et al, 2021). In brown adipocytes, RIIβ is the dominant form (Zhang et al, 2012), and mice deleted of RIIβ has been reported to be resistant to obesity, probably due to increased PKA activity (Czyzyk et al, 2008; Schreyer et al, 2001; Su et al, 2017). While downstream of $\beta_3$-AR-PKA signaling in thermogenesis is known (Kong et al, 2014; Sambeat et al, 2017; Sambeat et al, 2016; Yi et al, 2020), factors that may directly affect $\beta_3$-AR-PKA signaling, such as potential modification of PKA activity during thermogenesis, have not been studied.

Here we report identification of the first microprotein C16orf74 (MICT1) that is highly enriched in brown adipose tissue and is induced upon cold exposure. We show that C16orf74 that we named as MICT1 potentiates $\beta_3$-AR-PKA signaling by interacting with PP2B, prolonging PKA activity to promote thermogenesis. During submission of our current work for publication, others confirmed PP2B interaction with C16orf74 (Bradburn et al, 2004; Brauer et al, 2019). Furthermore, here we demonstrate in vivo impact of MICT1 on thermogenesis to affect adiposity and insulin sensitivity in mice. In fact, the GWAS database reveals a SNP in C16orf74 associated with type 2 diabetes. MICT1 may provide a new therapeutic target to increase thermogenesis and energy expenditure for obesity and obesity-related metabolic diseases, such as type 2 diabetes.

## Results

### MICT1 is a cold-induced microprotein highly enriched in brown adipocytes

C16orf74 (1190005I06Rik in mice) encodes a previously uncharacterized microprotein of 76 aa in length (8 kDa). There are a few reports on the human ortholog, C16orf74, as a marker for pancreatic without known function (Kushibiki et al, 2020; Nakamura et al, 2017). However, by RT-qPCR and immunoblotting, we found that C16orf74 was expressed highly in BAT compared to other tissues in mice (Figs. 1A and EV1A). We also detected C16orf74 in the adipocyte fraction of BAT, but not in the stromal vascular fraction (SVF) containing various other cell types including preadipocytes (Fig. 1B). Furthermore, C16orf74 mRNA levels were increased during brown adipocyte differentiation, similar to other BAT enriched genes, such as Pgc1α and Prdm16 (Figs. 1C and EV1B). We then examined C16orf74 expression in human beige adipocytes. Human adipose-derived mesenchymal stem cells were differentiated into white adipocytes by use of differentiation cocktail containing insulin, dexamethasone, and isobutylmethylxanthine (IBMX), followed by beiging via T3, rosiglitazone and forskolin treatment. As expected, these human beige adipocytes compared to white adipocytes expressed significantly higher levels of thermogenic genes, such as UCP1, DIO2, PGC1α, and CIDEA. More importantly, mRNA levels of human

ortholog of C16orf74 coding for MICT1 were induced by 8-fold upon beiging (Fig. 1D). Consistent with our observations, RNA-seq showed human C16orf74 ortholog to be higher in human beige adipocytes compared to preadipocytes or white adipocytes (Min et al, 2019) (Fig. EV1C). These results indicating enrichment of C16orf74 in thermogenic brown adipocytes and beige adipocytes suggested to us its potential function in thermogenesis, and we named this uncharacterized microprotein encoded by C16orf74 as Microprotein for Thermogenesis 1 (MICT1).

Since important BAT enriched genes that play key roles in thermogenesis are induced upon cold exposure to increase thermogenesis, we tested whether cold exposure or stimulation of $\beta_3$-AR-PKA signaling pathway induces expression of MICT1. Indeed, MICT1 mRNA levels in brown adipose tissue were increased by 4-fold when mice were exposed to cold at 4 °C than at thermoneutrality at 30 °C. Probably due to browning, MICT1 mRNA and protein levels were also increased in subcutaneous inguinal WAT (iWAT) of mice exposed to cold (Fig. 1E). In addition, MICT1 mRNA levels were increased when BAT cells (from S.Kajimura) were differentiated into brown adipocytes in culture and treated with the physiological ligand of $\beta_3$-AR norepinephrine, a $\beta_3$-AR specific agonist CL-316,243, or forskolin, an activator of adenylyl cyclase to increase cAMP levels (Fig. 1F). In this regard, we found three cAMP response elements (CREs), 5′-TGACGTCA-3′, at the 5′ upstream of the coding region of the MICT1 gene. Indeed, MICT1 promoter-reporter assays upon co-transfection of MICT1 promoter-reporter construct with CREB showed a 14-fold increase in luciferase activity (Figs. 1G and EV1D). Moreover, the mutation of CREB binding sites in this promoter-reporter assay resulted in a greatly decreased MICT1 promoter activity, further supporting CREB-mediated MICT1 induction during thermogenesis (Fig. 1G). Overall, we conclude that MICT1 is mainly expressed in thermogenic adipocytes and is induced upon cold via $\beta_3$-AR stimulation. Interestingly, GWAS database indicated a SNP in C16orf74 is associated with type 2 diabetes (https://www.ebi.ac.uk/gwas/genes/C16orf74), further suggesting its potential role in improving insulin sensitivity by brown adipose tissue thermogenesis.

### MICT1 overexpression promotes thermogenesis in brown adipocytes

To explore the potential role of MICT1 in brown adipocyte function of thermogenesis, we performed gain-of-function studies in brown adipocytes by measuring heat production and uncoupled oxygen consumption rates (OCR). First, we differentiated BAT cells into brown adipocytes by treating the confluent cells with differentiation cocktail containing insulin, dexamethasone, IBMX, indomethacin, T3, and rosiglitazone and then infected these cells with MICT1 adenovirus for overexpression (OE) or GFP adenovirus as a control. MICT1 overexpressing brown adipocytes had 5-fold higher MICT1 mRNA and protein levels (Fig. 2A, left, Fig. EV2A, left). To examine whether MICT1 modulates the thermogenic capacity, we utilized a small molecule thermosensitive fluorescent dye, ERthermAC, to directly monitor temperature changes in live cells (Kriszt et al, 2017). Heat production was quantified by the average percentage of ERthermAC+ cells in the region of higher temperature (R1). Via FACS, we detected a shift in distribution of control BAT to a higher temperature range when

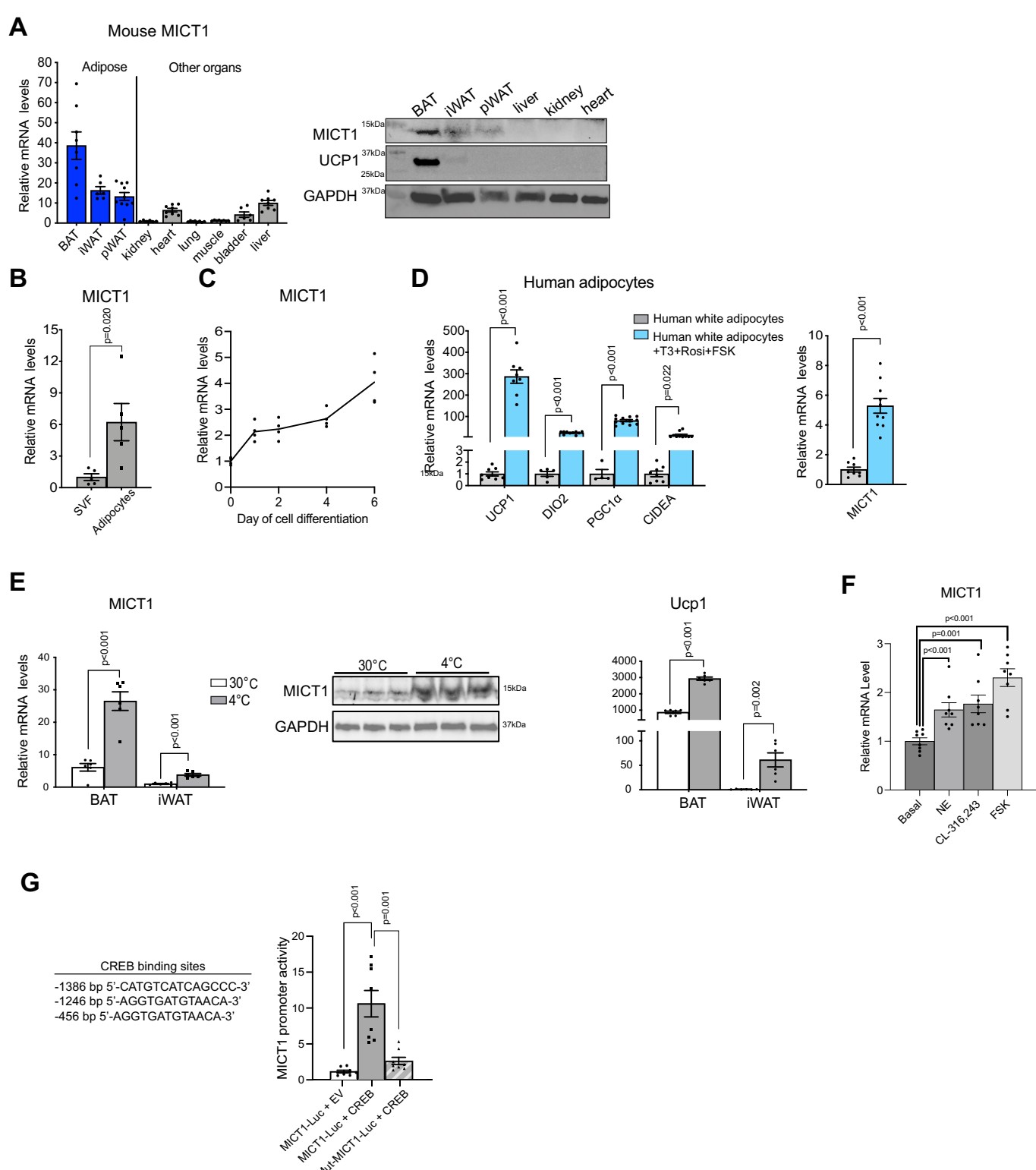

stimulated with forskolin to induce thermogenesis. More importantly, the average percent of ERthermAC⁺ MICT1 overexpressing cells in the higher temperature R1 compared to control cells was higher by approximately 20% and 30% in the basal and stimulated conditions, respectively (Fig. 2A, right), demonstrating that MICT1

overexpression resulted in higher heat production. Next, to test whether MICT1 overexpression affects mitochondrial respiration, we measured OCR by Seahorse in basal and CL-316,243-treated conditions. Indeed, MICT1 overexpressing cells compared to GFP control cells had higher OCR in all conditions, oligomycin-, FCCP-,

◀ **Figure 1. MICT1 is a microprotein highly enriched in brown adipocytes.**

(A) RT-qPCR ($n = 6$–8) and IB of MICT1 in various tissues from 12-wk-old C57BL/6 mice. (B) MICT1 mRNA levels in the adipocyte fraction and SVF of BAT ($n = 5$). (C) RT-qPCR for MICT1 during BAT cell differentiation ($n = 4$). (D) RT-qPCR for indicated genes in differentiated human white adipocytes and beige adipocytes treated with T3, rosiglitazone, and forskolin (FSK) (UCP1: $P < 0.0001$, DIO2: $P < 0.0001$, PGC1α: $P < 0.0001$, MICT1: $P < 0.0001$). (E) RT-qPCR of MICT1 and Ucp1 in BAT and iWAT from mice housed at either 30 °C or 4 °C ($n = 6$) and IB of MICT1 of BAT from mice housed at 30 °C or 4 °C ($n = 3$) (BAT MICT1: $P < 0.0001$, IWAT MICT1: $P = 0.0001$, BAT Ucp1: $P < 0.0001$, IWAT Ucp1: $P = 0.0018$). (F) MICT1 mRNA levels in differentiated BAT cells treated with norepinephrine (NE), CL-316,243 or FSK for 6 h ($n = 8$, NE: $P = 0.0014$, CL-316,243: $P = 0.0014$, FSK: $P < 0.0001$). (G) HEK293FT cells were transfected with −1.6 kb MICT1 promoter-luciferase reporter or MICT1 mutant promoter-luciferase reporter, in which CRE is mutated along with EV or CREB ($n = 8$, MICT1-Luc+CREB: $P < 0.0002$, Mut-MICT1 + CREB: $P = 0.0009$). Data is expressed as means ± standard errors of the means (SEM) of indicated number of biological replicates. The statistical differences in mean values were assessed by Student's *t* test. See also Fig. EV1. Source data are available online for this figure.

and rotenone/antimycin A-treated conditions (Fig. 2B, left). Importantly, in CL-316,243 treated condition, MICT1 overexpressing cells showed a significantly higher uncoupled OCR (Fig. 2B, right). We also measured OCR in these cells upon forskolin treatment and found that, similar to CL-316,243 treatment, MICT1 overexpressing cells compared to control cells had increased OCR (Fig. EV2A). After observing increased OCR, we next examined expression of various thermogenic genes. Differentiated BAT cells overexpressing MICT1 had higher Ucp1 mRNA levels by more than 3-fold in the stimulated condition. Expression of Pgc1α, Tfam, Nrf1, Dio2 was higher as well (Fig. 2C). In contrast, in differentiated 3T3-L1 adipocytes which were not treated with beiging agents and thus did not undergo beiging, MICT1 overexpression was not sufficient to induce thermogenic genes (Fig. 2D). These results, overall, demonstrate that MICT1 increases thermogenic activity and thermogenic genes in cultured brown adipocytes with thermogenic capacity.

## MICT1 ablation suppresses thermogenesis in brown adipocytes

Next for loss-of-function studies, we infected differentiated brown adipocytes with lentivirus containing the shRNA targeting MICT1 for knockdown (MICT1 KD). MICT1 knockdown cells had significantly lower MICT1 expression at both RNA and protein levels by RT-qPCR and immunoblotting, respectively (Fig. 3A). To test the functional outcome of MICT1 knockdown, we measured OCR and detected lower OCR in all conditions and notably lower uncoupled OCR in MICT1 knockdown cells compared to scrambled lentivirus infected control BAT cells, especially in the CL-316,243-stimulated condition (Fig. 3A). Furthermore, MICT1 knockdown in these differentiated brown adipocytes resulted in lower expression thermogenic genes, such as Ucp1, Pgc1α, Nrf1, Elovl3, and CideA (Fig. 3B).

Next, we employed CRISPR-Cas9 system to generate MICT1 knockout pools of BAT cells that stably expressing Cas9 (MICT1 KO) using two single guide RNAs (sgRNAs). The two independent knockout pools showed more than 95% lower MICT1 mRNA and protein levels compared to control BAT-Cas9 cells transduced with scrambled sgRNA (Fig. 3C, left, Fig. EV3D, left). Importantly, the use of ERthermAC in these cells upon differentiation showed that MICT1 knockout shifted the cell population to the lower temperature region compared to control cells by 30% and 50% in basal and stimulated conditions, respectively (Fig. 3C, right), indicating that MICT1 gene ablation decreases heat production in differentiated BAT cells. By seahorse assay, we also detected that MICT1 knockout cells had decreased OCR in all conditions. In CL-

316,243-treated condition, MICT1 knockout cells compared control cells showed a significantly lower uncoupled OCR (Fig. 3D). MICT1 knockout cells treated with forskolin also had lower OCR compared to controls cells (Fig. EV3D). MICT1 knockout cells treated with forskolin had significantly lower expression of thermogenic and mitochondrial biogenesis markers, including Ucp1, Pgc1α, Tfam, Nrf1 and Cox1 at both mRNA and protein levels (Fig. 3E). However, expression of these thermogenic genes was not unaffected at the basal condition. Additionally, mitochondrial content of MICT1 knockout cells, as determined by mitochondrial DNA/genomic DNA ratio and MitoTracker staining, were significantly lower than Scr cells upon stimulation (Fig. EV3B,C). To document the global impact of MICT1 on BAT cells, we performed RNA-seq in differentiated MICT1 knockout brown adipocytes. Notably, downregulated genes include those for thermogenesis, such as Ucp1 those for mitochondrial biogenesis and oxidative phosphorylation, such as Nrf1, Tfam (Dempersmier et al, 2015; Yi et al, 2020). In addition, genes important for insulin signaling, such as Irs1 and Akt2, and G-coupled receptor (GPCR) signaling pathways, such as Mapk14 (also known as p38α MAPK), Gene Ontology analysis of at least 2-fold downregulated genes in MICT1 ablated cells indicated that GPCR signaling pathway, lipid metabolic processes, mitochondrion organization, thermogenesis, regulation of brown adipocytes, and insulin signaling (Fig. EV3E) were significantly affected. The RNA-seq data is congruent with the above RT-qPCR in which MICT1 knockout cells stimulated with forskolin had significantly lower thermogenic and mitochondrial biogenesis markers. Unexpectedly, MICT1 knockout cells generated by CRISPR-Cas9 had lower expression of adipogenic genes, such as Pparγ, C/ebpβ, Fabp4 (Fig. EV3F,G). This was in contrast to the above MICT1 knockdown performed in differentiated brown adipocytes in which no changes in adipogenic markers, including Pparγ, Fabp4, and C/ebpβ, were detected (Fig. EV3A). It is plausible that due to the MICT1 deletion at the genomic level, MICT1 may have affected in vitro differentiation because of the use of common adipogenic cocktail containing IBMX to increase cAMP/PKA activity. Regardless, MICT1 is found mainly in brown adipocytes but not in SVF containing preadipocytes and cAMP/PKA function on adipogenesis in vivo, if any, is unclear. All together, we conclude that MICT1 promotes thermogenesis in brown adipocytes.

## Plasma membrane localization of MICT1 and its interaction with PP2B

In exploring how MICT1 promotes thermogenesis at the molecular level, we first examined MICT1 structure and its intracellular

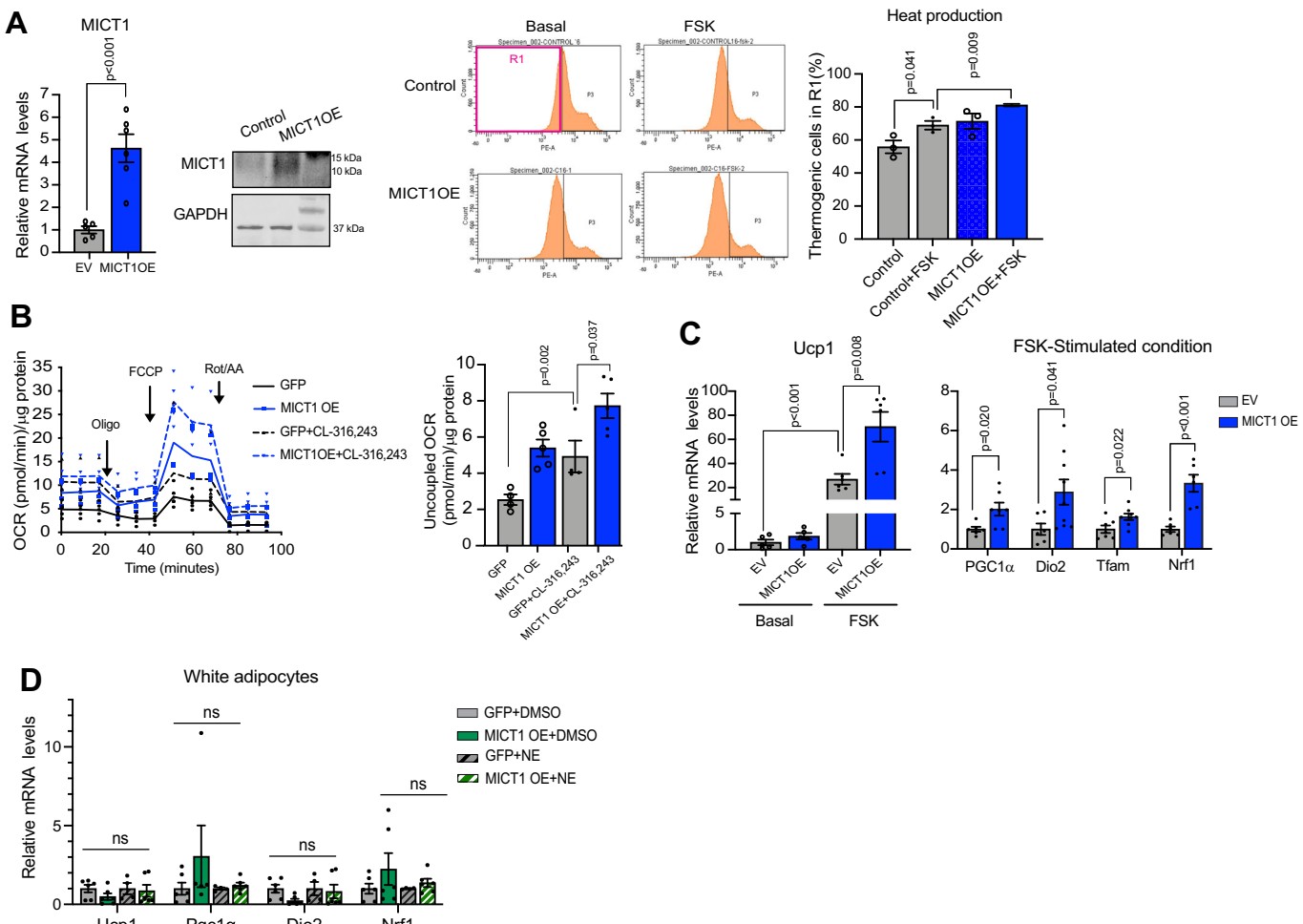

**Figure 2. MICT1 overexpression promotes on thermogenesis in cultured brown adipocytes.**

(A) (Left) RT-qPCR ($n = 5$, $P = 0.0006$) and IB for MICT1 in BAT cells transduced with GFP or MICT1 adenovirus for OE on Day 4 of differentiation. (Right) FACS analysis for ERthermAC for MICT1 OE BAT cells ($n = 3$). Region (R1) of higher temperature is boxed. (B) OCR and uncoupled OCR under oligomycin (0.5 μM), measured in MICT1OE BAT cells treated with CL-316,243 using Seahorse XFe24 analyzer ($n = 4$–5). (C) RT-qPCR for thermogenic in MICT1 OE BAT cells in the FSK-treated condition ($n = 4$–6, EV + FSK Ucp1: $P = 0.0005$, MICT1OE + FSK Ucp1: $P = 0.0005$, Nrf1: $P = 0.0004$). (D) RT-qPCR for thermogenic genes in 3T3-L1 adipocytes transduced with GFP or MICT1 adenovirus for OE on Day 4. Cells were treated with either DMSO or NE ($n = 4$–6). Data is expressed as means ± standard errors of the means (SEM) of indicated number of biological replicates. The statistical differences in mean values were assessed by Student's $t$ test. See also Fig. EV2.

localization. MICT1 and its human ortholog encode a 76 aa (8 kDa) microprotein (Fig. 4A). To investigate the intracellular localization of MICT1, we overexpressed C-terminal Flag-tagged mouse and human MICT1 isoforms in HEK293 cells. By subcellular fractionation and immunoblotting with Flag antibody, the 76 aa MICT1 was detected in the plasma membrane fraction (Fig. 4B, left). Similarly, immunofluorescence imaging detected the 76 aa MICT1 to be detected in plasma membrane, co-localizing with the plasma membrane marker, Na$^+$, K$^+$-ATPase (Fig. 4B, right, Fig. EV4A). When we overexpressed the human ortholog, which corresponds to the mouse 76 aa MICT1 in HEK293 cells, human MICT1 was also found in the plasma membrane fraction (Fig. 4C). In fact, MICT1 has an N-terminal glycine, the amino acid residue that can undergo myristoylation to anchor proteins to the membrane. To test whether this myristoylation site is important for MICT1 localization to plasma membrane for its action, we generated a MICT1 construct with G2A mutation. HEK293 cells were transfected with

G2A-MICT1 and cell lysates were subjected to subcellular fractionation. Immunoblotting showed that, unlike WT MICT1, without the myristoylation site, G2A-MICT1 was detected in the cytosolic fraction, which was also corroborated by immunofluorescence imaging (Figs. 4D and EV4A). Interestingly, bioinformatic analysis indicated the presence of a longer variant (111 aa,12 kDa) with 35 aa-extension at N-terminus via an alternate transcription start site. Unlike the main 76 aa MICT1 form, the 111 aa MICT1 lacked the N-terminal myristoylation site and was not colocalized with Na$^+$, K$^+$-ATPase upon overexpression in HEK293 cells, but it was detected only in the cytosol (Figs. 4E and EV4A).

We next examined the localization of endogenous MICT1 in brown adipose tissue from mice, by using MICT1 antibody that we generated against C-terminal peptides (DELGSYQDDGELEPEA) of MICT1. Immunoblotting using our MICT1 antibody detected the 76 aa MICT1, as well as 111 aa MICT1 upon subcellular fractionation of brown adipose tissue from mice. The dominant 76

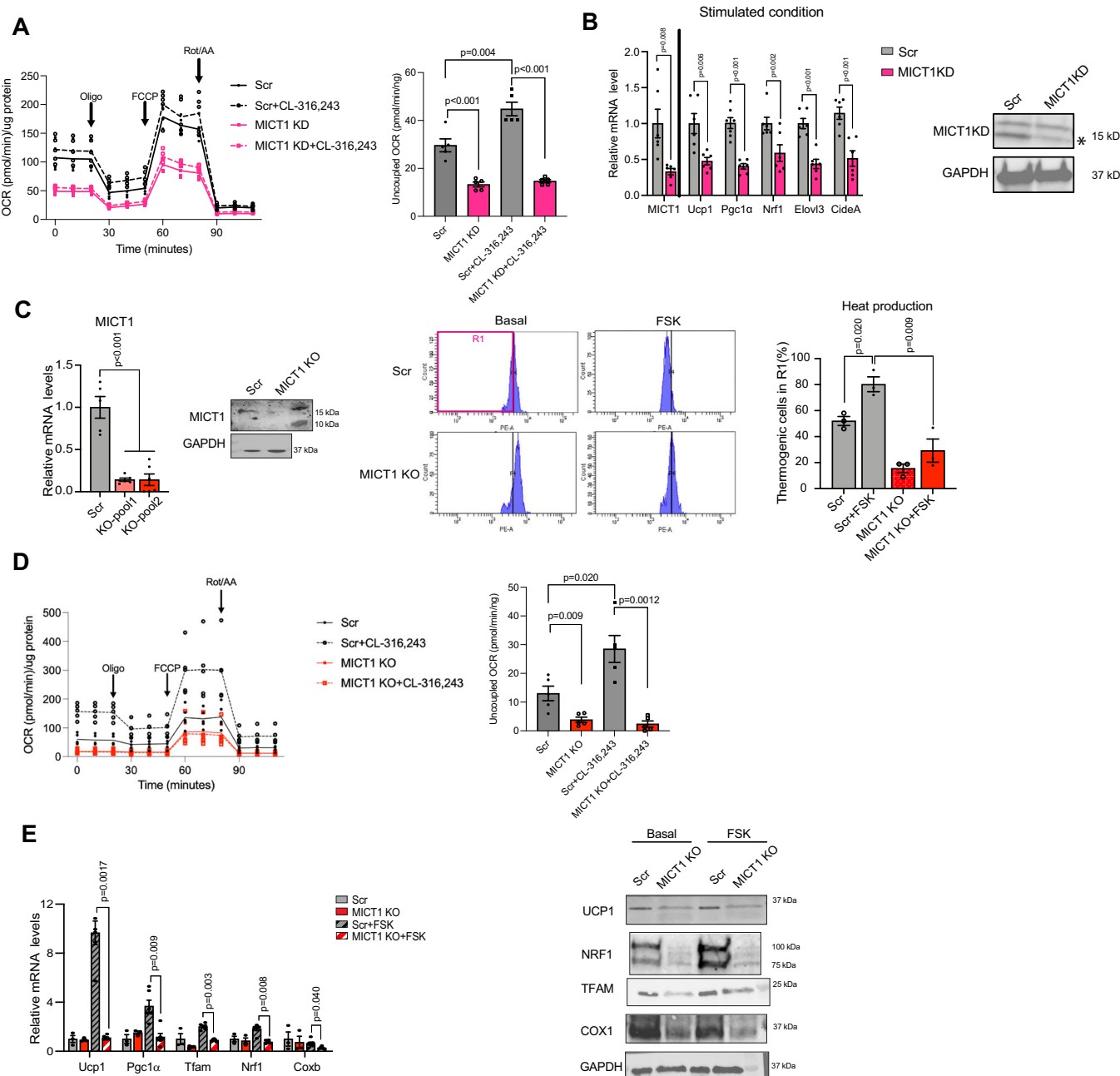

**Figure 3. MICT1 ablation suppresses thermogenesis in cultured brown adipocytes.**

(A) OCR measured in MICT1 KD treated with CL-316,243 using Seahorse XFe24 analyzer, and relative uncoupled OCR under oligomycin (0.5 μM) ($n = 5$, MICT KD: $P = 0.0005$, MICT1 KD + CL-316,243: $P < 0.0001$). (B) (Left) RT-qPCR for thermogenic genes in MICT1 KD BAT cells in the FSK-treated condition ($n = 6$, Pgc1α: $P = 0.0001$, Elovl3: $P = 0.0001$, CideA: $P = 0.0009$). (Right) IB for MICT1 in MICT1 KD BAT cells. (C) (Left) RT-qPCR ($n = 5–6$, KO-pool1: $P < 0.0001$, KO-pool2: $P = 0.0002$) and IB for MICT1 in Scr or MICT1 KO. (Middle) FACS analysis for ERthermAC for MICT1 KO ($n = 3$). Region (R1) of higher temperature is boxed. (Right) Quantification of number of thermogenic cells in R1. (D) OCR measured in MICT1 KO treated with CL-316,243 using Seahorse XFe24 analyzer, and relative uncoupled OCR under oligomycin (0.5 μM) treatment ($n = 4–5$, MICT1 KO + CL-316,243: $P = 0.0012$). (E) (Left) RT-qPCR for indicated genes in Scr or MICT1 KO in the basal ($n = 3$) and the FSK-stimulated condition ($n = 6$, Ucp1: $P = 0.0017$, Pgc1α: $P = 0.0093$, Tfam: $P = 0.0031$, Nrf1 = 0.0083, Cox8b: $P = 0.0402$). (Right) IB for UCP1, NRF1, TFAM, and COX1 in Scr or MICT1 KO. Data is expressed as means ± standard errors of the means (SEM) of indicated number of biological replicates. The statistical differences in mean values were assessed by Student's $t$ test. See also Fig. EV3.

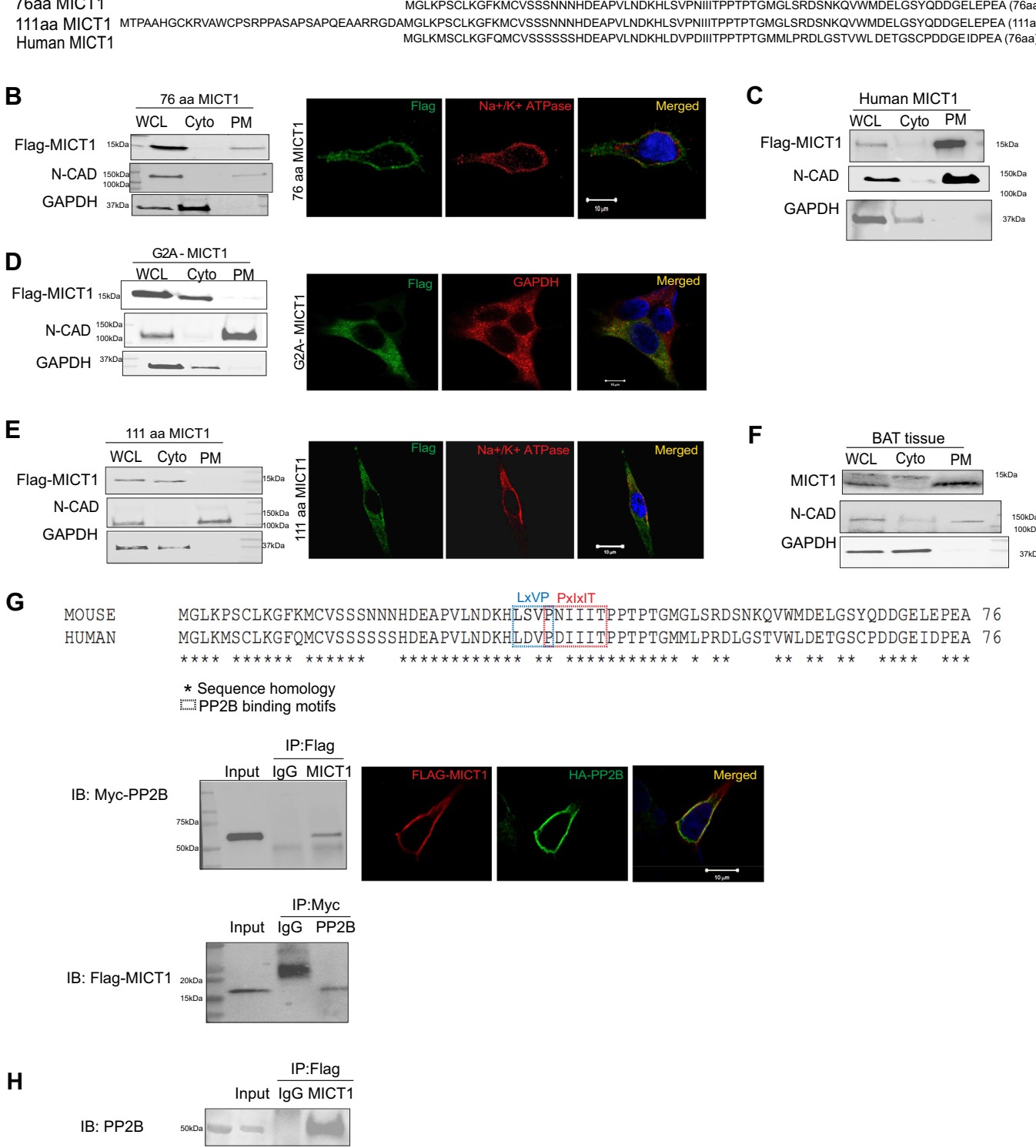

**A**

76aa MICT1    MGLKPSCLKGFKMCVSSSNNNHDEAPVLNDKHLSVPNIIITPPTPTGMGLSRDSNKQVWMDELGSYQDDGELEPEA (76aa)
111aa MICT1  MTPAAHGCKRVAWCPSRPPASAPSAPQEAARRGDAMGLKPSCLKGFKMCVSSSNNNHDEAPVLNDKHLSVPNIIITPPTPTGMGLSRDSNKQVWMDELGSYQDDGELEPEA (111aa)
Human MICT1             MGLKMSCLKGFQMCVSSSSSSHDEAPVLNDKHLDVPDIIITPPTPTGMMLPRDLGSTVWLDETGSCPDDGEIDPEA (76aa)

aa MICT1 was indeed found in the plasma membrane fraction, whereas the minor 111 aa MICT1 was found in the cytosol and at a greatly lower level than the dominant 76 aa MICT1 form (Figs. 4F and EV4B). These results confirmed that the 76 aa MICT1 is the main form of MICT1 present in brown adipose tissue of mice and

in humans and is localized at the plasma membrane. Thus, our present study is on this major 76 aa MICT1.

Since MICT1 belongs to the microprotein class which generally regulates biological processes by affecting protein-protein interaction, we examined the potential structural motifs of MICT1.

**Figure 4. Plasma membrane localization of MICT1 and its interaction with PP2B.**

(A) Primary amino acid sequence of mouse (76 aa) and human MICT1 (76 aa). N-terminal 35aa extension of MICT1 variant is present in mice (111aa MICT1). (B) (Left) HEK 293FT cells transfected with 76 aa form of C-terminal Flag-tagged MICT1. Whole cell lysates (WCL), cytosol (Cyto), or plasma membrane (PM) fractions subjected to IB. (Right) Immunofluorescence (IF) staining of HEK293 cells expressing MICT1-Flag (scale bar: 10 μm). (C) Human MICT1 transfected HEK 293FT cell lysates were fractionated and subjected to IB. (D) (Left) HEK293FT cells transfected with C-terminal Flag-tagged G2A-MICT1 mutant were fractionated and subjected to IB. (Right) IF staining of HEK293FT cells transfected with C-terminal Flag-tagged G2A-MICT1 mutant (scale bar: 10 μm). (E) (Left) HEK 293FT cells transfected with 111 aa form of C-terminal Flag-tagged MICT1 were fractionated and subjected to IB. (Right) IF staining of HEK293 cells expressing MICT1-Flag (scale bar: 10 μm). (F) (Left) BAT from C57BL/6 mice was fractionated and subjected to IB using antibody against C-terminal peptide of MICT1. (G) (Top) Primary amino acid sequences of mouse and human MICT1. Asterisks indicate the amino acid homology between mouse and human MICT1. aa sequence in boxes, (PxIxIT) and (LxVP), are the two consensus PP2B (also called CaN) binding sequences. (Bottom left) Myc-tagged PP2B and Flag-tagged MICT1 were transfected into HEK293 cells, and cell lysates were immunoprecipitated (IP) with Flag antibody followed by IB with Myc antibody and vice versa for a reverse CoIP. (Bottom right) IF staining showing MICT1 and PP2B colocalization at the plasma membrane of HEK293 cells (scale bar: 10 μm). (H) IP of MICT1 in brown adipose tissue from mice overexpressing Flag-tagged MICT1 followed by IB of PP2B. See also Fig. EV4.

Indeed, we found that both mouse and human MICT1 contain two consensus PP2B docking motifs (Li et al, 2011; Wigington et al, 2020): in mouse MICT1, PNIIIT and LSVP and in human MICT1, PDIIIT and LDVP (Fig. 4G, top). PP2B is a widely expressed serine/threonine phosphatase that acts on its protein substrates by the docking motifs PxIxIT and LxVP. Therefore, we performed co-immunoprecipitation to test the potential MICT1-PP2B interaction in HEK293 cells upon transfection with Myc-tagged PP2B and Flag-tagged MICT1. Immunoprecipitation of MICT1 followed by immunoblotting for PP2B, and vice versa, showed their interactions (Fig. 4G, bottom left). In addition, immunofluorescence imaging detected the 76 aa MICT1 to be colocalized with PP2B in HEK293 cells (Fig. 4G, bottom right). In addition, using brown adipose tissue from mice overexpressing Flag-tagged MICT1, immunoprecipitation of MICT1 followed by immunoblotting for PP2B shows their interaction in vivo (Fig. 4H). Overall, we conclude that MICT1 is a microprotein that is mainly found in the plasma membrane and can interact with PP2B.

## MICT1 potentiates PKA activity

So far, our study showed that MICT1 promotes thermogenesis in brown and in beige adipocytes in culture. We then detected MICT1 at the plasma membrane and MICT1 interaction with PP2B. In this regard, PP2B has been known to dephosphorylate RII at the plasma membrane. PP2B, through its interaction with a PKA binding protein, AKAP, allows RII dephosphorylation, causing its re-association with C subunit that limits PKA activation (Bock et al, 2021; Church et al, 2021). In fact, the dominant isoform of R subunit of PKA in BAT is RIIβ (Gold et al, 2011). Also, our RNA-seq analysis (Fig. EV3E) indicated that GPCR signaling pathway was the most downregulated pathway upon MICT1 ablation. Since MICT1 interacts with PP2B, which is known to dephosphorylate RII, and β3AR-cAMP-PKA is the critical signal pathway for thermogenesis, we tested whether MICT1 affects PKA activity. First, we overexpressed MICT1 in HEK293 cells that have minimal endogenous MICT1 and measured cAMP levels and PKA activity in cell extracts. As expected, treatment with forskolin increased cAMP levels in control cells. While MICT1 overexpression did not increase cAMP levels, PKA activity was greatly increased in cells overexpressing MICT1 (Fig. EV5A, left). H89, the PKA inhibitor, abolished PKA activity, verifying that the activity we measured was indeed PKA activity (Fig. EV5A, right). Similarly, MICT1 over-expressing cells compared to control cells had higher PKA activity

upon CL-316,243 treatment (Fig. 5A). So far, we measured PKA activity in cell extracts. Next, we examined the changes in PKA activity in live cells. We utilized an excitation-ratiometric PKA activity reporter, ExRai-AKAR2, which modulates the peak excitation wavelength of cpEGFP in response to phosphorylation by PKA in live cells (Zhang et al, 2021; Zhang et al, 2022). While there was no difference in PKA activity in differentiated BAT cells in the basal condition, upon norepinephrine treatment, MICT1 overexpressing cells showed higher PKA activity in live cells, with PKA activity increasing by a much greater magnitude than control cells (Fig. 5B). We then treated the cells with H89 at the final time points. H89 treated cells showed a significantly decreased PKA activity in live cells, indicating that observed cpEGFP live cell imaging indeed represented PKA activity.

Next, we employed MICT1 knockout cells to examine cAMP and PKA activity. MICT1 knockout cells had significantly lower PKA activity, especially in the stimulated condition, while there were no differences in cAMP levels between MICT1 knockout and control cells (Fig. EV5B). We also measured PKA activity in brown adipocytes that were treated with norepinephrine. As expected, both norepinephrine and CL-316-243 treatment of control BAT cells significantly increased PKA activity. Similarly, control cells treated with IBMX, a phosphodiesterase inhibitor to increase cAMP levels, also showed an increase in PKA activity. More importantly, norepinephrine, IBMX, or CL-316,243 treatment of MICT1 knockout cells resulted in significantly lower PKA activity when compared with control cells treated with the same agents (Fig. 5C). Furthermore, ratiometric PKA activity measurement in live MICT1 knockout cells, compared to the control cells, showed a severely lower PKA activity in the stimulated condition in these live cells (Fig. 5D).

As we detected clear evidence for the potentiation of PKA activity by MICT1, we next examined phosphorylation status of PKA targets, such as CREB (S133) and HSL (S563, S660). As expected, phosphorylation of these PKA targets was increased when differentiated BAT cells were treated with forskolin for 15 min. More importantly, MICT1 overexpression in BAT cells further increased phosphorylation of CREB (S133), HSL (S563, S660) and p38 (T180, Y182), upon forskolin stimulation (Figs. 5E and EV5D, left). Conversely, we detected significantly lower phosphorylation of HSL (S563, S660), CREB (S133), p38 (T180, Y182) in MICT1 knockout cells compared to control BAT cells in both basal and stimulated conditions, demonstrating the impact of MICT1 on phosphorylation of PKA target proteins in brown adipocytes upon

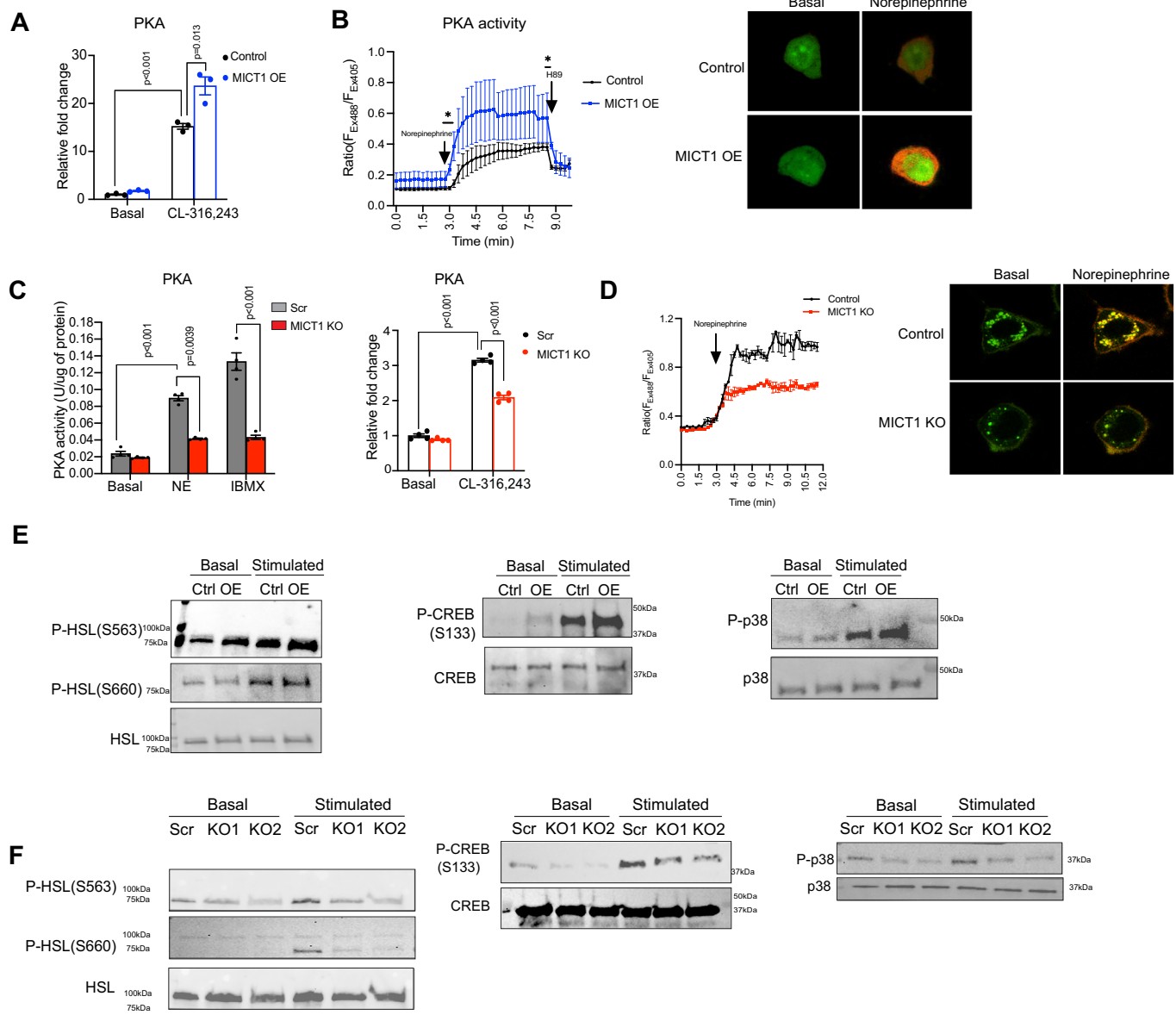

**Figure 5. MICT1 potentiates PKA activity.**

(A) PKA activity measured in the basal condition or after CL-316,243 treatment ($n = 3$, $P < 0.0001$). (B) Time course of PKA activity in live control and MICT1 OE HEK cells transfected with ExRai-AKAR2 plasmid ($n = 3$–4). (Right) Representative cell images of ExRai-AKAR2 response in the control or MICT1 OE cells in the basal or FSK-stimulated conditions. (C) PKA activity measured in MICT1 KO cells under various treatments: NE or IBMX treatments (Left, $n = 4$, Scr+NE: $P = 0.0002$, MICT1 KO + NE: $P = 0.0039$, MICT1 KO + IBMX: $P < 0.0001$) or CL-316,243 (Right) ($n = 4$, Control+CL-316,243: $P < 0.0001$, MICT1 KO + CL-316,243: $P < 0.0001$). (D) (Left) Time course of PKA activity in live Scr and MICT1 KO cells were transduced with ExRai-AKAR2 lentivirus ($n = 3$). (Right) Representative cell images of ExRai-AKAR2 response in Scr or MICT1 KO cells in the basal or FSK-stimulated conditions. (E) Lysates from differentiated control or MICT1 OE BAT cells in the basal and FSK-stimulated conditions were subjected to IB for P-HSL (S563 and S660), P-CREB (S133), and P-p38. (F) Lysates from Scr and MICT1 KO cells in the basal and FSK-stimulated conditions used for IB for P-HSL (S563 and S660), P-CREB (S133), and P-p38. Data is expressed as means ± standard errors of the means (SEM) of indicated number of biological replicates. The statistical differences in mean values were assessed by Student's t test. See also Fig. EV5.

MICT1 ablation (Figs. 5F and EV5D, right). To understand changes in protein phosphorylation at the global level, we also performed phosphoproteomic analysis using control Scr and MICT1 knockout brown adipocytes treated with forskolin. MICT1 ablation decreased phosphorylation of proteins downstream of PKA signaling, including PRKAR2B, PRKACA, and MAPK14. Gene Ontology and the pathway analysis also indicated that the major pathway

which was decreased most significantly by MICT1 ablation was the PKA pathway (Fig. EV5C, left). Specifically, we found significantly lower phosphorylation of PRKAR2B, PRKARCA, MAPK14, and MAP2K6, as well as other known PP2B downstream targets, including DNML1, GYS2, and MEF2C (Fig. EV5C, right). Furthermore, phosphoproteomic analysis using control and MICT1 overexpressing brown adipocytes resulted in increased

phosphorylation of those proteins known to be downstream targets of PKA signaling, such as PRKAR2B, PRKARCA, and HSL, as well as other PP2B downstream targets, including GSK3 and BCL2L13 (Fig. EV5D). Overall, our results demonstrate that MICT1 potentiates PKA activity to promote phosphorylation of PKA targets in brown adipocytes.

## Plasma membrane MICT1-PP2B interaction controls RIIβ dephosphorylation in potentiating PKA activity for thermogenesis

So far, we found that MICT1 potentiates PKA activity to promote thermogenesis. We also found that MICT1 interacts with PP2B at the plasma membrane (Fig. 4G). We next tested whether MICT1-PP2B interaction affects RIIβ phosphorylation for MICT1 potentiation of PKA activity to promote thermogenesis.

First, we examined whether PP2B is required for MICT1 potentiation of PKA activity by using small molecular inhibitor of PP2B, FK506. To this end, we treated BAT cells with FK506 and measured PKA activity in the presence or absence of MICT1. As expected, we observed considerably higher PKA activity in MICT1 overexpressing brown adipocytes compared to control cells stimulated with forskolin treatment. More importantly, when the cells were pretreated with FK506, the enhanced PKA activity in forskolin-stimulated MICT1 overexpression was mitigated (Fig. 6A, left), demonstrating that MICT1 functions through PP2B. Similarly, PP2B knockdown using shRNA lentivirus targeting PP2B also mitigated the increase in CL-316,243 stimulated PKA activity upon MICT1 overexpression (Fig. EV6A). Moreover, in MICT1 overexpressing BAT cells, we detected increased RIIβ phosphorylation by immunoblotting using Phospho-specific RIIβ (S112) antibody (Fig. 6A, right). However, this enhanced RIIβ phosphorylation was mitigated upon knockdown of PP2B in MICT1 overexpressing cells (Fig. EV6B). Additionally, when human MICT1 was overexpressed in human beige adipocytes, PKA activity was significantly higher than that in control beige adipocytes. This increase in PKA activity was abolished in cells pretreated with FK506 (Fig. 6b). Conversely, MICT1 knockout cells compared to control BAT cells showed lower PKA activity upon stimulation, and FK506 pretreatment or PP2B KD prevented this decrease in PKA activity (Figs. 6C and EV6D). Immunoblotting for Phospho-RIIβ showed significantly decreased PKA RIIβ phosphorylation in MICT1 knockout cells in both basal and the forskolin-stimulated conditions (Fig. 6C, middle, Fig. EV6D), while total and Phospho-PKA RIIα levels remained unchanged (Fig. EV6E). Next, to examine whether MICT1 ablation affects the PP2B-RIIβ interaction, we used MICT1 knockout cells and control cells to perform co-immunoprecipitation. After immunoprecipitation with PP2B, we detected higher RIIβ protein levels in MICT1 knockout cells, demonstrating that MICT1 ablation increases RIIβ-PP2B interaction, thereby increasing dephosphorylation of RIIβ (Fig. 6c, right, Fig. EV6f). These results demonstrate the requirement of PP2B for the MICT1's impact on PKA activity by affecting PKA RIIβ phosphorylation.

Although we clearly demonstrate MICT1 functions by interacting with PP2B, given the potential that a transcript could function not only as an mRNA but also as a regulatory RNA, we inserted a stop codon at the beginning of MICT1 gene (Stop codon mutant) to test the possibility of MICT1 functioning as an RNA for brown adipocyte thermogenesis. We overexpressed MICT1 Stop codon mutant and MICT1 in differentiated BAT cells. When treated with forskolin, BAT cells expressing MICT1 Stop codon mutant showed significantly lower Ucp1 mRNA levels compared to MICT1 overexpressing cells (Fig. 6D, left). In addition, while MICT1 overexpression clearly showed increased Phospho-PKA RIIβ upon forskolin treatment, MICT1 Stop codon mutant overexpression did not, verifying that potentiation of PKA activity by MICT1 was not due to MICT1 RNA acting as a regulatory RNA (Fig. 6D, right, Fig. EV6I).

Next, to examine whether membrane localization of MICT1 is required for its potentiation of PKA activity, we compared PKA activity in BAT cells expressing MICT1 and myristoylation-defective G2A-MICT1. Indeed, unlike plasma membrane bound MICT1, cytosolic G2A-MICT1 did not affect PKA activity nor could G2A-MICT1 increase heat production or OCR (Fig. 6E–G). Thus, myristoylation of MICT1 and thus plasma membrane localization is required for its promotion of thermogenesis. Next, as we found above that MICT1 can interact with PP2B and, since PP2B substrates commonly employ one of the two motifs, PxIxIT and LxVP, to dock PP2B, we examined which one of the PP2B binding motifs of MICT1 is necessary for the MICT1-PP2B interaction. We mutated each of the putative PP2B interaction domains, PNIIIT to PNAAAT (Mutant 1) and LSVP to LAVA (Mutant 2). HEK293 cells were transfected with Myc-PP2B and MICT1, or MICT1 mutants. Co-immunoprecipitation showed that the MICT1-PP2B interaction was significantly ablated when PNIIIT motif of MICT1 was mutated, while LSVP motif mutation still showed strong interaction, indicating that MICT1 interacts with PP2B through its PNIIIT motif (Fig. 6H). Moreover, immunoblotting showed that, while MICT1 and MICT1 Mutant 2 had comparable Phospho-RIIβ, MICT1 Mutant 1 had greatly lower Phospho-RIIβ (Figs. 6I and EV6G). In addition, we tested whether binding to PP2B via PNIIIT motif affects thermogenesis as well as PKA RIIβ phosphorylation in differentiated brown adipocytes. We measured OCR to test the functional outcome of the reduced binding to PP2B. Overall, we detected reduced OCR only by MICT1 Mutant 1, but not MICT1 Mutant 2 (Fig. 6J). Overall, these results show MICT1 interacts with PP2B via its PNIIIT motif for modulation of PKA activity through affecting RIIβ phosphorylation in promotion of thermogenesis.

We next employed INCA-6 that specifically acts on the PxIxIT motif and blocks at the substrate recognition site (Roehrl et al, 2004) by administering in mice in vivo. We administered either saline or INCA-6 to high-fat diet (HFD) fed mice via direct injection into brown adipose tissue (Fig. 6K, left). At 4 °C, INCA-6-treated mice had higher core body and brown adipose tissue temperatures, demonstrating significantly improved thermogenic capacity, as well as increased OCR in brown adipose tissue (Fig. 6K,L). Moreover, INCA-6 injected mice had significantly increased RIIβ phosphorylation (Fig. 6M, Fig. EV6H, left). Because PP2B is known to dephosphorylate NFAT to increase transcription of inflammatory genes, we examined and found that PP2B inhibition by INCA-6 did not alter expression of inflammatory factors in differentiated brown adipocytes (Fig. EV6H, right).

All together, we conclude that MICT1 functions as a microprotein by localizing at the plasma membrane for interaction with PP2B through its PNIIIT motif to potentiate PKA activity by affecting phosphorylation of RIIβ to promote thermogenesis.

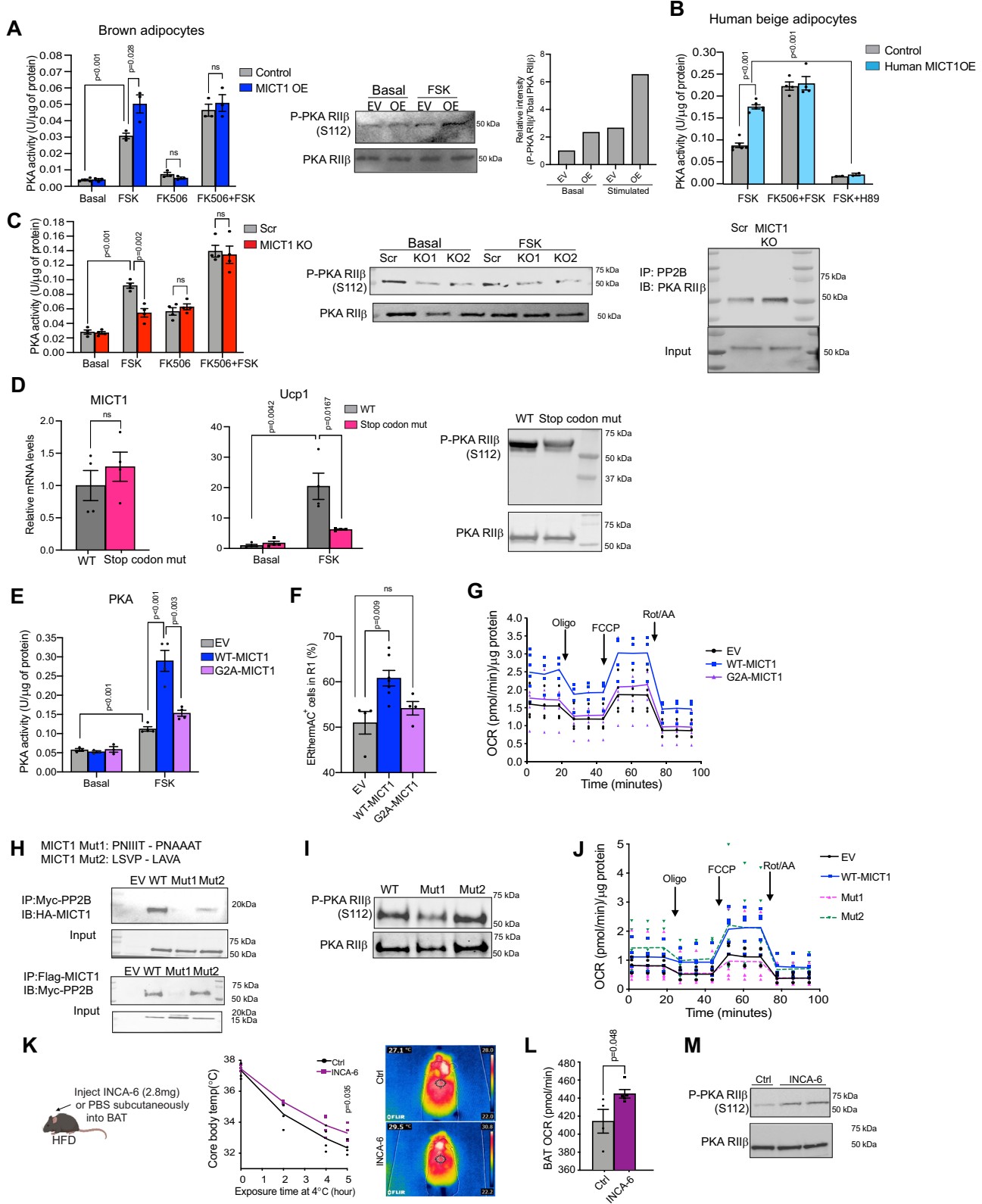

**Figure 6.  Plasma membrane MICT1-PP2B interaction controls RIIβ dephosphorylation for potentiation of PKA activity and thermogenesis.**

(A) Control or MICT1 OE BAT cells were treated with FK506, a PP2B inhibitor. (Left) PKA activity was measured in the basal and FSK-treated conditions ($n = 3$, Control +FSK: $P < 0.0001$). (Right) IB of MICT1 OE BAT cells using antibody for P-PKA RIIβ (S112). (B) MICT1 OE human beige adipocytes were treated with FK506. PKA activity was measured in the basal and FSK-treated conditions ($n = 4$, MICT1 OE + FSK: $P < 0.0001$, MICT1 OE + FSK + H89: $P < 0.0001$). (C) Scr or MICT1 KO cells treated with FK506. (Left) PKA activity was measured in the basal and FSK-treated conditions ($n = 4$, Scr FSK: $P < 0.0001$). (Middle) IB of MICT1 KO cells using P-PKA RIIβ (S112) antibody. (Right) IB for PKA RIIβ after IP using PP2B antibody in MICT1 KO cell lysates. (D) (Left) RT-qPCR for MICT1 and Ucp1 in HEK293FT cells transfected with MICT1 that had a stop codon inserted at the beginning of MICT1 gene ($n = 4$, WT + FSK Ucp1: $P = 0.0042$, Stop codon mut+FSK: $P = 0.0167$). (Right) IB for P-PKA RIIβ of HEK293FT cells overexpressing WT-MICT1 or MICT1 Stop codon mut. (E) BAT cells transfected with either EV, WT-MICT1 or G2A-MICT1 mutant constructs were differentiated to brown adipocytes. On Day 6, lysates were used to measure PKA activity in the basal and FSK-treated conditions ($n = 4$, EV + FSK: $P = 0.0007$, WT + FSK: $P = 0.0007$). (F) FACS analysis for ERthermAC in BAT cells transfected with either EV, WT-MICT1 or G2A-MICT1 ($n = 4$). (G) OCR measured in BAT cells overexpressing WT-MICT1 and G2A-MICT1 by using Seahorse XFe24 Analyzer. (H) Two consensus PP2B binding sequences of MICT1 (PNIIIT and LSVP) were mutated to PNAAAT and LAVA, respectively. IB using Flag antibody for MICT1 after IP with Myc antibody for PP2B using lysates from HEK293FT cells transfected with Myc-PP2B and various Flag-MICT1 mutant constructs. IP with Flag antibody for MICT1, followed by IB with Myc antibody for PP2B for a reverse CoIP. (I) IB for P-PKA RIIβ of BAT cells overexpressing WT-MICT1 or Flag-MICT1 mutant constructs. (J) OCR measured in BAT cells overexpressing WT-MICT1, Mut1 or Mut2 by using Seahorse XFe24 Analyzer. (K) Experimental design of INCA-6 injection study. Core body temperature measured at 4 °C at indicated time points (hrs) ($n = 4$). Infrared thermography of BAT after 4 h of cold exposure at 4 °C. (L) OCR measured in BAT using Seahorse XFe24 Analyzer ($n = 4$). (M) IB of INCA-6 injected BAT lysates using P-PKA RIIβ (S112) antibody. Data is expressed as means ± standard errors of the means (SEM) of indicated number of biological replicates. The statistical differences in mean values were assessed by Student's $t$ test. See also Fig. EV6.

## Overexpression of MICT1 in brown adipose tissue in mice promotes thermogenesis, mitigating obesity and insulin resistance

In order to demonstrate physiological significance of MICT1, we generated MICT1 transgenic mice harboring the LoxP-STOP-LoxP cassette inserted at the 5' of MICT1 to overexpress by UCP1 Cre-mediated excision of the stop codon. In these mice overexpressing MICT1 in brown adipose tissue (MICT1-BSOE), MICT1 mRNA levels were higher by 6-fold in brown adipose tissue but not in other tissues (Fig. 7A). MICT1 protein levels also were higher significantly in brown adipose tissue of MICT1-BSOE mice (Figs. 7B and EV7A). MICT1-BSOE mice compared to wild type littermates had increased expression of UCP1 at mRNA and protein levels (Figs. 7B and EV7B). Other thermogenic genes, such as Tfam, Pgc1α, Cox8b, Elovl3, and CideA were also induced in BAT, while expression of adipogenic markers, such as Pparγ, Fabp4, C/ebpβ, and AdipoQ were not affected (Fig. EV7A). We next subjected these mice to acute cold exposure at 4 °C to examine their thermogenic capacity. Upon 3 h of cold challenge, MICT1-BSOE mice had higher core body temperature by 3 °C compared to wild-type littermates (Fig. 7C, left). The differences in core body temperature were even greater after 4 h of cold challenge. Indeed, upon cold exposure, brown adipose tissue of MICT1-BSOE mice displayed about 2 °C higher temperature than that of WT littermates detected by infrared thermography (Fig. 7C, right). Similarly, female MICT1-BSOE mice exhibited significantly higher cold tolerance with higher UCP1 protein levels (Fig. EV7C). Considering the increased thermogenic capacity in MICT1-BSOE mice, we explored the role of MICT1 on energy balance by indirect calorimetry using CLAMS. The MICT1-BSOE mice had significantly higher OCR during nights at room temperature (23 °C) and cold ambient temperature (4 °C) than wild type littermates (Fig. 7D). We next tested whether changes in respiratory activity in whole body OCR of these mice was contributed by changes in energy expenditure from brown adipose tissue. Brown adipose tissue excised from MICT1-BSOE mice had significantly higher OCR, as measured by Seahorse assay, evidence of contribution by brown adipose tissue to the enhanced whole-body metabolic activity (Fig. 7E).

Reflecting enhanced energy expenditure, MICT1-BSOE mice were leaner than control wild-type littermates even on a chow diet and after 7 weeks (wks) and lighter by 4 g compared to the WT littermates on HFD (Fig. 7F, left). The body composition analysis by EchoMRI showed that MICT1-BSOE mice had smaller fat mass without changes in lean mass, and the weight gain was primarily accounted by adipose accumulation (Fig. 7F, right). Upon HFD, MICT1-BSOE mice had smaller lipid droplets in brown adipose tissue compared to WT, in which more than nearly 60% of lipid droplets were less than 200 μm², probably for more efficient lipolysis to release fatty acids for binding to UCP1 and fueling thermogenesis in the mitochondria. These mice also had increased UCP1 protein levels shown by whole mount immunostaining of BAT (Fig. 7G). In addition, blood glucose levels of MICT1-BSOE mice were significantly lower during glucose tolerance test (GTT), and they had improved insulin sensitivity by insulin tolerance test (ITT) (Fig. 7H). Overall, these results demonstrate that over-expression of MICT1 in Ucp1⁺ cells protects these mice from diet-induced obesity and insulin resistance via increasing thermogenesis and energy expenditure.

Since MICT1 increases thermogenesis, we also tested whether MICT1 can rescue insulin resistance and adiposity of genetically obese mice using adeno-associated viral (AAV) vectors. Ob/ob mice at 10 wks of age were administered intravenously with $5 \times 10^{11}$ viral genomes of serotype 8 AAV vectors encoding either GFP or MICT1 coding sequence, whose expression was under the control of the UCP1 promoter (AAV8-GFP or AAV8-UCP1-MICT1). Following AAV administration, mice were maintained on chow followed by 4 wks of HFD, and body weights and metabolic measurements were monitored. AAV8-UCP1-MICT1 treated mice showed overexpression of MICT1 by mRNA and protein levels in BAT, while other tissues, such as WAT and liver, were not affected (Fig. 7I). They also had significantly higher Ucp1 mRNA and protein levels, as well as other thermogenic genes (Fig. 7I). Furthermore, AAV8-UCP1-MICT1 mice showed higher core body temperature and brown adipose tissue temperature detected by infrared thermography imaging of brown adipose tissue region when exposed to 4 °C (Fig. 7J). Their whole-body OCR by CLAMS also was higher upon cold exposure, while not significantly different at ambient temperature and thermoneutrality (Fig. 7K, left). Indeed, the

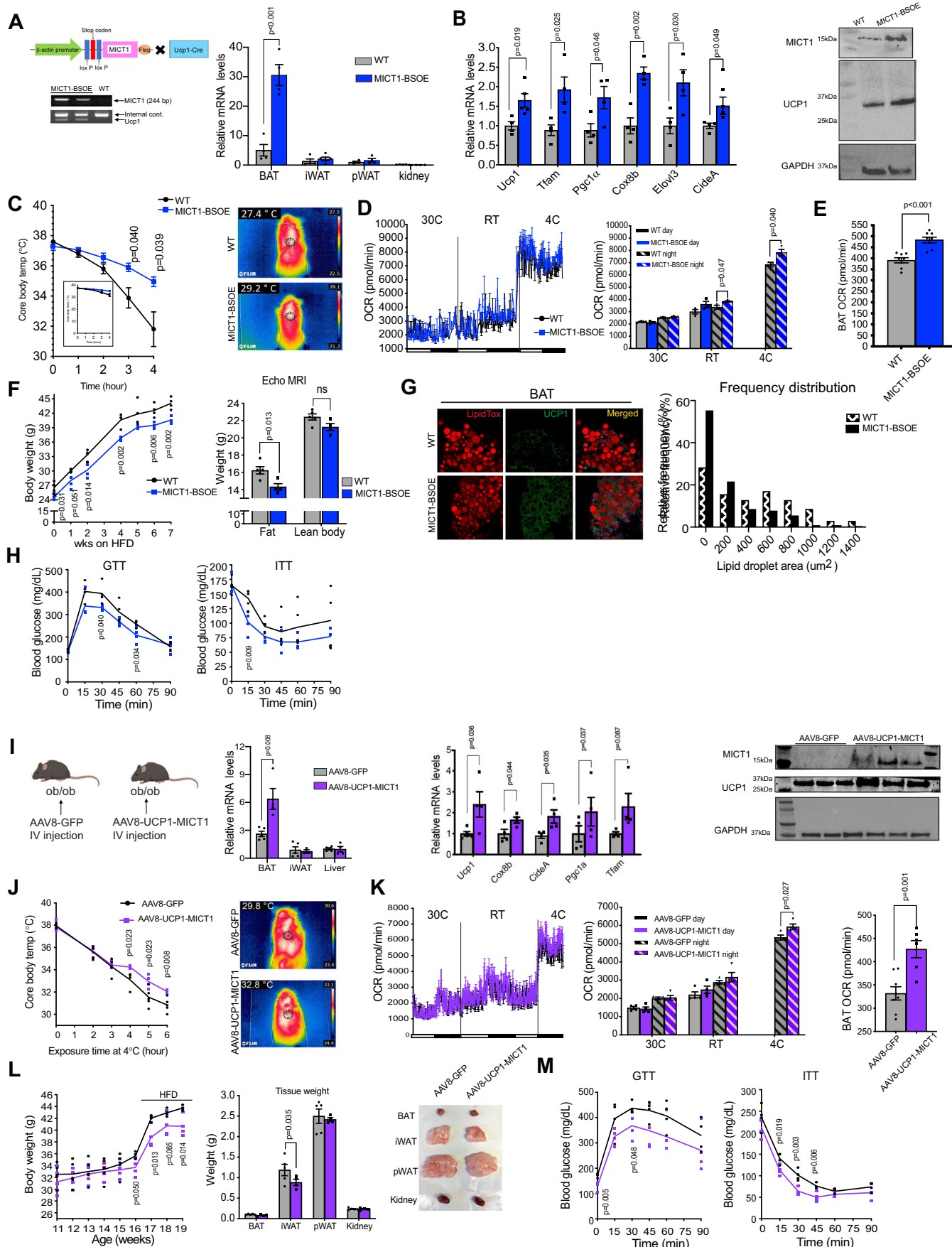

**Figure 7. MICT1 overexpression in BAT in mice promotes thermogenesis, preventing obesity and insulin resistance.**

(A) (Left) Schematic diagram of the strategy used to generate BAT-specific MICT1 conditional OE mice. PCR genotyping of the mice: Top, MICT1 allele; bottom, UCP1-Cre. (Right) RT-qPCR for MICT1 in BAT, iWAT, pWAT, and kidney from WT and MICT1-BSOE mice ($n = 4$, BAT: $P < 0.0001$). (B) (Left) RT-qPCR for thermogenic genes in BAT from MICT-BSOE mice ($n = 4$). (Right) IB for MICT1 and UCP1 in BAT from MICT1-BSOE mice. (C) Core body temperature measured in 14-wks-old mice at 4 °C at indicated time points ($n = 5$). Infrared thermography of BAT after 6 h of cold exposure at 4 °C. (D) Whole body VO$_2$ assayed in WT and MICT1-BSOE mice, housed at indicated ambient temperatures by indirect calorimetry using CLAMS ($n = 5$). (E) OCR measured in BAT of WT and MICT1-BSOE mice using Seahorse XFe24 Analyzer ($n = 8$, $P = 0.0001$). (F) Body weights of WT and MICT1-BSOE mice on HFD for 7 wks and body composition assessed by EchoMRI ($n = 4$). (G) (Left) Whole-mount immunostaining of UCP1 (green) and LipidTox (red) in BAT of 24-week-old mice on HFD for 8 wks. (Right) Frequency distribution of lipid droplet areas in BAT. (H) GTT and ITT of WT and MICT1-BSOE mice on HFD ($n = 4$). (I) (Left) Schematic diagram of the experiment design that 10-wk-old ob/ob mice injected intravenously with control, AAV8-GFP or AAV8-UCP1-MICT1 for MICT1 OE in UCP1$^+$ cells. Mice were on HFD starting at 16.5 wks-old. (Right) RT-qPCR for MICT1 in BAT, iWAT, and liver of AAV8-UCP1-MICT1 mice and control mice ($n = 4$), RT-qPCR for thermogenic genes in BAT, and IB for MICT1 and UCP1 in BAT from mice ($n = 4$). (J) Core body temperature measured at 4 °C at indicated time points ($n = 4$). Infrared thermography of BAT after 6 h of cold exposure at 4 °C. (K) (Left) Whole body VO$_2$ assayed in AAV8-UCP1-MICT1 mice and control mice housed at indicated temperatures by indirect calorimetry using CLAMS ($n = 4$). (Right) OCR measured in BAT using Seahorse XFe24 Analyzer. (L) Body weights and tissue weights of AAV8-UCP1-MICT1 mice and control mice ($n = 4–5$). (M) GTT and ITT of AAV8-UCP1-MICT1 mice and control mice on HFD ($n = 4$). Data is expressed as means ± standard errors of the means (SEM) of indicated number of biological replicates. The statistical differences in mean values were assessed by Student's $t$ test. See also Fig. EV7.

enhanced energy expenditure upon cold exposure was contributed by significantly higher OCR from brown adipose tissue, measured by Seahorse assay using excised brown adipose tissue pieces (Fig. 7K, right). With increased thermogenic capacity and energy expenditure, AAV8-UCP1-MICT1 ob/ob mice were significantly leaner by more than 3 g than the control mice, with significantly smaller iWAT mass especially upon HFD feeding (Fig. 7l). In addition, blood glucose levels of AAV8-UCP1-MICT1 ob/ob mice were significantly lower during GTT and had improved insulin sensitivity by ITT (Fig. 7M). Together, we conclude that MICT1 promotes thermogenesis and enhances energy expenditure in mice and can reduce adiposity and improve insulin sensitivity in obesity.

## MICT1 ablation in brown adipose tissue in mice reduces thermogenic capacity to gain adiposity and insulin resistance

To further examine the role of MICT1 in vivo, we performed MICT1 ablation in UCP1$^+$ cells in mice. We first generated MICT1 floxed mice by inserting two LoxP sites flanking exon 2 of MICT1 via CRISPR-Cas9 system. Germline transmitted floxed mice were crossed with UCP1-Cre mice to generate Ucp1$^+$ cell ablation of MICT1 in mice (MICT1-BSKO). In addition to genotyping, we also confirmed that MICT1 mRNA levels were lower by 80% in brown adipose tissue (Fig. 8A). Similarly, MICT1 protein levels were significantly lower in brown adipose tissue of MICT1-BSKO mice, which was accompanied by a 60% reduction in UCP1 mRNA levels and other thermogenic genes. UCP1 protein levels also were drastically reduced in brown adipose tissue of MICT1-BSKO mice (Fig. 8B). However, we did not detect significant differences in expression of adipogenic markers, such as Pparγ, Fabp4, C/ebpβ, and AdipoQ (Fig. EV8A).

Next, to study the physiological outcome from decreased UCP1 and other thermogenic gene expression, we subjected the MICT1-BSKO mice to acute cold exposure at 4 °C. After 5 h of cold exposure, the MICT1-BSKO mice were significantly less cold-intolerant compared to control mice (Fig. 8C, left). Moreover, the brown adipose tissue of MICT1-BSKO mice displayed markedly lower temperature compared to that of control mice upon 6 h of cold exposure, assessed by infrared thermography (Fig. 8C, right). These results clearly demonstrate that MICT1-BSKO mice had impaired thermogenic capacity with lower UCP1 expression in

brown adipose tissue. We further assessed the functional consequences of loss of MICT1 in UCP1$^+$ cells on energy balance. Indeed, the MICT1-BSKO mice showed a significantly lower OCR compared to control mice at all temperatures at night and showed an even greater OCR reduction at 4 °C, while locomotive activity and food intake were similar (Fig. 8D, left). By using Seahorse XFe24, we found a significantly lower OCR in brown adipose tissue isolated from MICT1-BSKO mice, an indication of brown adipose tissue contribution to changes in whole-body energy expenditure (Fig. 8D, right). Reflecting the impairment of thermogenesis and energy expenditure, MICT1-BSKO mice displayed significantly higher body weight compared to control mice on HFD. The body composition analysis by EchoMRI showed that MICT1-BSKO mice had increased fat mass without changes in lean mass (Fig. 8E), demonstrating that the weight gain was primarily accounted by adipose accumulation. In addition, MICT1-BSKO mice on HFD had significantly higher blood glucose levels during glucose tolerance test and had impaired insulin sensitivity (Fig. 8F). We obtained similar results with female MICT1-BSKO mice (Fig. EV8A–D). Further, MICT1 KO compared to WT mice showed a greatly lower RIIβ phosphorylation in brown adipose tissue (Fig. 8G). We conclude that MICT1 is essential for robust thermogenesis by inducing UCP1 and other target genes, likely through potentiated PKA activity.

Since we found MICT1 promotes thermogenesis and enhances insulin sensitivity in mice, we next investigated effect of MICT1 in human beige adipocytes capable of thermogenesis. Cells derived from human mesenchymal stem cells that were differentiated into adipocytes were treated with beiging agents. These beige adipocytes were transduced with human MICT1 lentivirus to test whether MICT1 overexpression can enhance thermogenic capacity. Indeed, MICT1 overexpression induced *UCP1* gene expression by 8-fold compared to control empty vector transduced cells. Expression of other thermogenic genes, such as *ELOVL3*, *COX7A*, and *PGC1α*, was also significantly higher upon MICT1 overexpression (Fig. 8H). Further, MICT1 overexpressing human beige adipocytes exhibited increased glucose uptake rate (Fig. 8I). Immunoblotting showed significantly increased Phospho-Akt (Fig. 8J). Conversely, human beige adipocytes transduced with lentivirus containing the shRNA targeting human MICT1 greatly lowered glucose uptake rate (Fig. 8K). Together, these data show that MICT1 promotes thermogenic gene expression and can improve insulin sensitivity in human beige adipocytes.

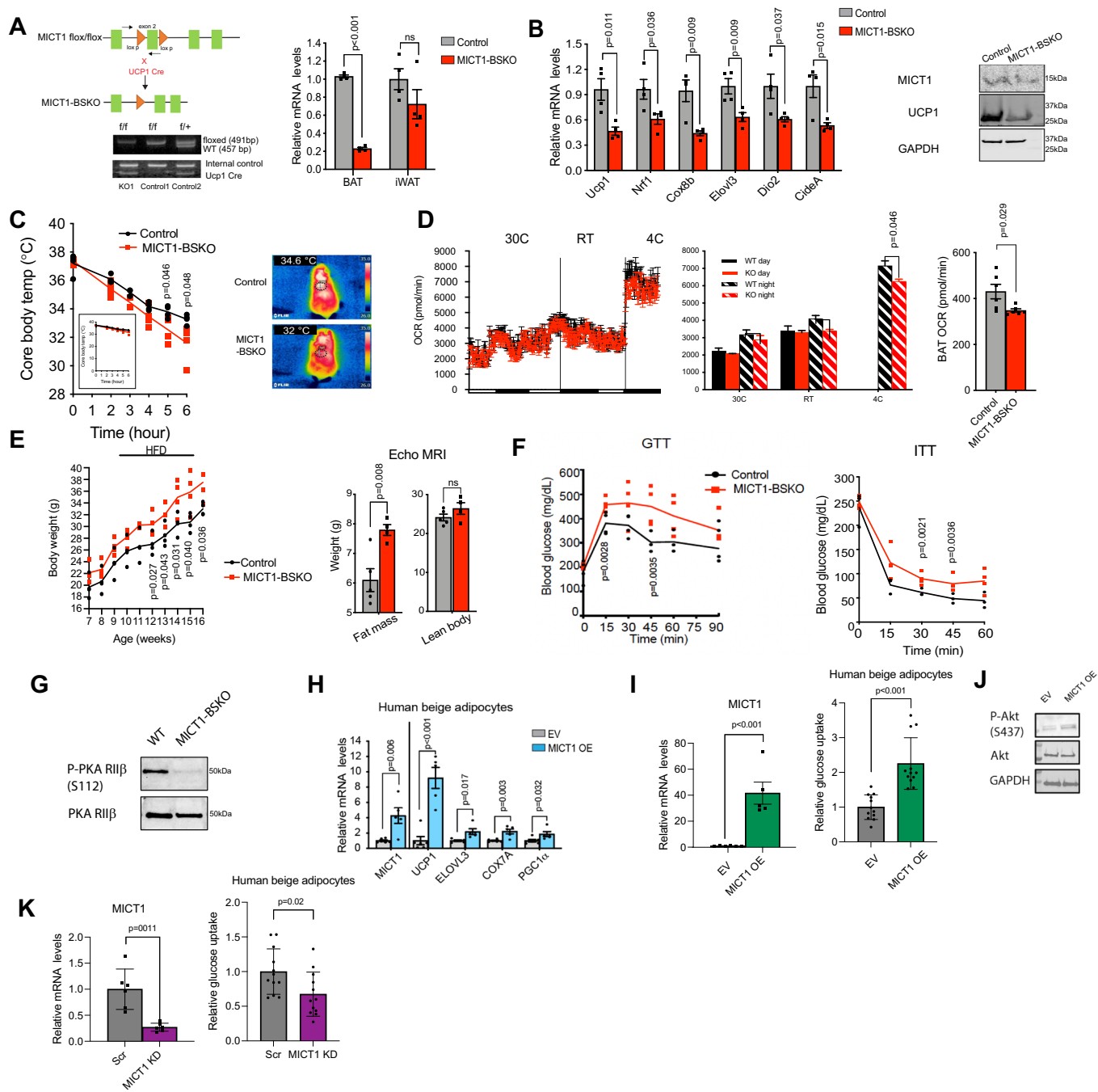

**Figure 8. MICT1 ablation in BAT in mice reduces thermogenic capacity to gain adiposity.**

(A) (Left) Schematic diagram of the strategy used to generate BAT-specific MICT1 conditional KO mice. PCR genotyping of the mice: Top, MICT1 allele and floxed allele; bottom, UCP1-Cre. (Right) RT-qPCR for MICT1 in BAT and iWAT from MICT1-BSKO and control mice ($n = 4$, BAT: $P < 0.0001$). (B) (Left) RT-qPCR for thermogenic genes in BAT from MICT-BSKO mice ($n = 4$). (Right) IB for MICT1 and UCP1 in BAT from MICT1-BSKO mice. (C) Core body temperature measured in 13-wks-old mice at 4 °C at indicated time points ($n = 4$). Infrared thermography of BAT after 3 h of cold exposure at 4 °C. (D) (Left) Whole body VO$_2$ assayed in WT and MICT1-BSKO mice housed at indicated temperatures by indirect calorimetry using CLAMS ($n = 4$). (Right) OCR measured in BAT of control and MICT1-BSKO mice using Seahorse XFe24 Analyzer. (E) Body weights of control and MICT1-BSKO mice on HFD from 10-wks old and body composition assessed by EchoMRI ($n = 4$). (F) GTT and ITT of MICT1-BSKO mice and control littermates on HFD ($n = 4$). (G) IB of P-PKA-RIIβ and total PKA RIIβ of BAT lysates from WT or MICT1-BSKO mice. (H) RT-qPCR for indicated genes in human beige adipocytes transduced with EV or human MICT1 lentiviruses for OE on Day 4 of differentiation ($n = 5$, UCP1: $P = 0.0002$). (I) RT-qPCR of MICT1 ($n = 5$, $P < 0.0001$) and glucose uptake rate of human beige adipocytes transduced with EV or human MICT1 lentiviruses ($n = 12$, $P < 0.0001$). (J) IB of EV or MICT1 OE human beige adipocyte lysates using P-Akt and total Akt antibodies. (K) RT-qPCR of MICT1 ($n = 6$, $P = 0.0011$) and glucose uptake rate of human beige adipocytes transduced with Scr or human MICT1 shRNA lentiviruses ($n = 12$). Data is expressed as means ± standard errors of the means (SEM) of indicated number of biological replicates. The statistical differences in mean values were assessed by Student's *t* test. See also Fig. EV8.

## Discussion

Microproteins are a rapidly expanding class of peptides/small proteins that utilize short sequences of 2–4 aa to bind large protein complexes and regulate biological processes. Although they make up a sizable fraction of the proteome, there are no known microproteins that affect thermogenesis. We found C16orf74 (MICT1) to be an unknown microprotein most highly expressed only in brown adipose tissue, compared to all other tissues, a depot known to be associated with improvements in insulin sensitivity. Notably, GWAS database indicated that MICT1 has SNPs associated with Type 2 diabetes. However, C16orf74 has been described to be elevated in pancreatic cancer, but no function in thermogenesis has been reported (Kushibiki et al, 2020; Nakamura et al, 2017). While downstream factors of the $\beta_3$-AR-PKA signaling pathway have been well studied for activation of thermogenesis during β3-adrenergic stimulation and cold-exposure, there is paucity of information on how $\beta_3$-AR-PKA signaling may be modulated during thermogenesis. Here, we identify for the first time a microprotein critical for thermogenesis. We demonstrate MICT1 (C16orf74) as a cold-induced microprotein that is specifically found in brown adipocytes and interacts with PP2B to potentiate PKA activity to promote thermogenesis.

Our study demonstrates that MICT1 interacts with PP2B to increase PKA activity by augmenting phosphorylation and expression of PKA downstream targets for thermogenesis. Here, we dissected the specific step at which MICT1 impacts $\beta_3$-AR-PKA signaling. cAMP binding to R subunit of PKA holoenzyme causes release and activation of C. The RII isoform can be dephosphorylated by phosphatases, such as PP2B, to enable the recapture of C to limit PKA activity. RII is bound to the plasma membrane or other subcellular membranes via PKA Anchoring Proteins (AKAPs) that provide for localization and temporal regulation in the formation of cAMP-PKA signalosomes (Ikeda et al, 2017; Zaccolo et al, 2021). While PP2B is best known to dephosphorylate NFAT to allow its translocation to the nucleus, PP2B has various other substrates, including RII, in a broad spectrum of cellular processes (Blumenthal et al, 1986; Cummings et al, 1996). It has been shown previously in neuronal cells that the tandem anchoring of PP2B and RII to AKAP5 allows dephosphorylation of RII by PP2B to limit PKA activity (Church et al, 2021; Gold et al, 2011). Here, we demonstrate that in brown adipocytes, MICT1 sustains PKA pathway signaling by binding and sequestering PP2B for disruption of PP2B-AKAP interaction, preventing dephosphorylation of RIIβ (Zhang et al, 2015). Moreover, transcriptional activation of MICT1 by CREB, a downstream transcription factor of PKA, demonstrates that MICT1 functions as an amplifier of $\beta_3$-AR-PKA signaling during cold exposure for promotion and maintenance of thermogenesis. In this regard, sarcoplasm/ER $Ca^{2+}$ ATPase 2b (SERCA2b)-RyR2 and adrenergic signaling can increase $Ca^{2+}$ cycling and $Ca^{2+}$ can stimulate phosphatase activity of PP2B (Chen et al, 2017; Guarnieri et al, 2022; Ikeda et al, 2017). However, for prolonged thermogenesis, we show that MICT1 induced upon cold exposure functions to prevent PP2B mediated dephosphorylation of RIIβ to sustain PKA activity.

While both mouse and human MICT1 contain the two consensus docking motifs for PP2B, PxIxIT and LxVP, mutation of only the PNIIIT binding domain of MICT1, and not LSVP domain, disrupts MICT1 interaction with PP2B. Thus, we found that MICT1 interacts with PP2B specifically through the PNIIIT docking motif in brown adipocytes. In fact, a recent preprint by Bradburn et al also indicated that C16orf74, which they term calcimembrin, can bind to PP2B by the

PxIxIT motif (Bradburn et al, 2004). Some of the other known PP2B interacting partners also have the two binding motifs in close proximity, but PP2B does not bind both but rather either one or the other, due to the large distance between the PP2B binding pockets for PxIxIT and LxVP motifs (Cyert). Further, Brauer et al recently has identified potential PP2B substrates by using an unbiased and systematic approach of in vivo Short linear motifs (SLiMs) dependent proximity labeling with in silico modeling (Brauer et al, 2019). Indeed, MICT1 was predicted to be one such PP2B interactor. Moreover, by administrating the PP2B chemical inhibitor INCA-6 that specifically targets the PxIxIT motif (Roehrl et al, 2004) into mice, we demonstrate that inhibition of PP2B via PxIxIT suppresses RIIβ dephosphorylation and promotes thermogenesis.

We uncovered two variants of MICT1: a 111 aa MICT1 found in the cytosol and a 76 aa MICT1 at the plasma membrane. Subcellular localization can provide different chemical environments and interacting partners and thus may confer different functionality of MICT1 to form complexes with different substrates. In our study, with the use of the N-terminus-specific primers for the 111 aa MICT1, as well as antibodies that recognize the C-terminal region of both forms, we discovered that the 76 aa MICT1 is the predominant form present in brown adipose tissue. Further studies can explore how expression of two forms is regulated to allow potential MICT1 interaction with interacting partners in different tissues.

Ablation of MICT1 in Ucp1[+] cells in mice impairs thermogenic capacity with lower OCR and promotes weight gain and insulin resistance. Conversely, overexpression of MICT1 in Ucp1[+] cells in mice brought higher thermogenic gene expression, with enhanced cold tolerance and energy expenditure. Thus, these mice remain leaner even on high fat diet with smaller lipid droplets in brown adipose tissue, exhibiting improved glucose tolerance and insulin sensitivity. Altogether, MICT1 is protective against diet-induced and genetic obesity and insulin resistance by increasing thermogenesis via its interaction with PP2B to potentiate PKA activity. In this regard, GWAS database showed that MICT1 gene (C16orf74) has two SNPs (rs377457 and rs439967) located in the intronic regions that are associated with Type 2 diabetes. We demonstrate here that AAV-mediated MICT1 administration to diet-induced or genetically obese mice can decrease adiposity and insulin resistance, revealing its therapeutic potential in obesity and insulin resistance. Moreover, small molecule inhibitors, such as INCA-6, that specifically act on the PxIxIT motif to block the substrate recognition site may represent potential therapeutics as they can mimic MICT1 to affect RIIβ phosphorylation to prolong PKA activity for thermogenesis to increase energy expenditure.

## Methods

**Reagents and tools table**

| Reagent/Resource | Reference or Source | Identifier or Catalog Number |
|---|---|---|
| **Experimental models** | | |
| 3T3-L1 | UCB Cell Culture Facility | N/A |
| HEK293FT | UCB Cell Culture Facility | N/A |

| Reagent/Resource | Reference or Source | Identifier or Catalog Number |
|---|---|---|
| BAT | Dr. Shingo Kajimura, Harvard University | N/A |
| Adipose-derived Mesenchymal Stem Cells; Normal, Human | ATCC | PCS-500-011 |
| AAVPro293T | Takara | 632273 |
| **Recombinant DNA** | | |
| pcDNA3.1(+)-ExRai-AKAR2 | Addgene | 161753 |
| pcDNA3.1 +/C-(K)DYK MICT1 | GenScript | |
| Human MICT1 | Origene | RC21684 |
| **Antibodies** | | |
| MICT1 | This laboratory | N/A |
| UCP1 | Abcam | 10983 |
| GAPDH | Cell Signaling Technology (CST) | 14C10 |
| FLAG | Cell Signaling Technology (CST) | 14793 |
| P-CREB | Cell Signaling Technology (CST) | 9198S |
| Total CREB | Cell Signaling Technology (CST) | 9197S |
| P-P38 | Cell Signaling Technology (CST) | 4511 |
| Total P38 | Cell Signaling Technology (CST) | 8690 |
| P-HSL (S563) | Cell Signaling Technology (CST) | 4139 |
| P-HSL (S660) | Cell Signaling Technology (CST) | 45804 |
| Total HSL | Cell Signaling Technology (CST) | 18381S |
| N-CAD | Cell Signaling Technology (CST) | 13116 |
| Pan-Calcineurin A | Cell Signaling Technology (CST) | 2614 |
| MYC | Sigma-Aldrich | C3956 |
| P-PKA RIIβ | Sigma-Aldrich | SAB4301261 |
| Total PKA RIIβ | Sigma-Aldrich | SAB4502353 |
| HA | Sigma-Aldrich | H3683 |
| Na+/K+ ATPase | Novus | NB300-146 |

| Oligonucleotides and other sequence-based reagents | Forward | Reverse |
|---|---|---|
| MICT1 TG | CTC CTG TCT GAA AGG CTT TAA G | CTG GAA CAT CGT ATG GGT AC |
| MICT1 Flox | GAG TGC GGC TCA GTG GTA TAG C | GGC TGA AGC CTC CCT GTC TG |
| UCP1 Cre | CCA TCT GCC ACC AGC CAG | TCG CCA TCT TCC AGC AGG |
| Internal Control | ACT GGG ATC TTC GAA CTC TTT GGA C | GAT GTT GGG GCA CTG CTC ATT CAC C |

| Reagent/Resource | Reference or Source | Identifier or Catalog Number |
|---|---|---|
| MICT1 | CCT GGT GTC CCT CAC GG | GCT TCA GGT TCC AGC TCT CC |
| Ucp1 | ACT GCC ACA CCT CCA GTC ATT | CTT TGC CTC ACT CAG GAT TGG |
| Tfam | GGC AAA GGA TGA TTC GGC TC | CAC TTC GTC CAA CTT CAG CC |
| Nrf1 | GCA CCT TTG GAG AAT GTG GT | CTG AGC CTG GGT CAT TTT GT |
| Cidea | TGT TCT TCT GTA TCG CCC AGT | GCC GTG TTA AGG AAT CTG CTG |
| Dio2 | CAG TGT GGT GCA CGT CTC CAA TC | TGA ACC AAA GTT GAC CAC CAG |
| Pgc1α | TAT GGA GTG ACA TAG AGT GTG CT | CCA CTT CAA TCC ACC CAG AAA G |
| Elovl3 | TCC GCG TTC TCA TGT AGG TCT | GGA CCT GAT GCA ACC CTA TGA |
| Cox8b | TGT GGG GAT CTC AGC CAT AGT | AGT GGG CTA AGA CCC ATC CTG |
| Cox7a | CAG CGT CAT GGT CAG TCT GT | AGA AAA CCG TGT GGC AGA GA |
| 16S | CTA GAA ACC CCG AAA CCA AA | CCA GCT ATC ACC AAG CTC GT |
| Human MICT1 | GTC CTG AAA GTC AAG CAC CTG | GAA GTT CTT GTT GGT GCT TAT GG |
| Human UCP1 | CCA ACT GTG CAA TGA AAG TGT | CAA GTC GCA AGA AGG AGG GTA |
| Human DIO2 | CCT CCT CGA TGC CTA CAA AC | GCT GGC AAA GTC AAG AAG GT |
| Human PGC1α | CCA CAG AGA ACA GAA ACA GCA | TGG GGT CAG AGG AAG AGA TAA |
| Human CIDEA | ACT CTG GAT GCC CTC GTC AT | ACT CTT CTG TGT CCA CCA CG |
| Human ELOVL3 | CTG TTC CAG CCC TAT AAC TTC | GAA TGA GGT TGC CCA ATA CTC C |
| Human COX7A | AGC GAA TTG GCA CCA AAG CAG CA | CTG GTG GCT CTG CCT TGC CAT |

| Chemicals, Enzymes and other reagents | Reference or Source | Identifier or Catalog Number |
|---|---|---|
| LipidTOX Red Reagent | ThermoFisher | H34476 |
| CL-316,243 | Abcam | AB144605 |
| Forskolin | Sigma-Aldrich | F3917 |
| Norepinephrine | Sigma-Aldrich | A7257 |
| Inca-6 | MedChem Express | HY-108544 |
| Insulin | Millipore-Sigma | I9278 |
| 3-Isobutyl-1-methylxanthine, IBMX | Sigma-Aldrich | I5879 |
| Indomethacin | Sigma-Aldrich | I7378 |
| Dexamethasone | Sigma-Aldrich | D2915 |
| T3 | Sigma-Aldrich | T6397 |
| Rosiglitazone | Sigma-Aldrich | R2408 |
| FK506 | Sigma-Aldrich | F4679 |
| H89 | Sigma-Aldrich | B1427 |

| Reagent/Resource | Reference or Source | Identifier or Catalog Number |
|---|---|---|
| Oligomycin | Sigma-Aldrich | 75351 |
| Carbonyl cyanide 4-(trifluoromethoxy) phenylhydrazone, FCCP | Sigma-Aldrich | C2920 |
| Rotenone | Sigma-Aldrich | R8875 |
| Antimycin A | Sigma-Aldrich | A8674 |
| **Software** | **Reference or Source** | **Identifier or Catalog Number** |
| Partek | PartekFlow | N/A |
| **Other** | **Reference or Source** | **Identifier or Catalog Number** |
| Colorimetric ELISA cAMP Kit | Abcam | ab234585 |
| PKA Colorimetric Activity Kit | ThermoFisher | EIAPKA |
| Dual-Glo Luciferase Kit | Promega | E2920 |
| Glucose Uptake-Glo Assay | Promega | J1341 |

## Methods and protocols

### Animals

All animal studies were carried out in accordance with University of California at Berkeley ACUC and OLAC regulations. Conditional MICT1 overexpressing mice and MICT1-floxed mice were produced by UC Berkeley Gene Targeting Facility. MICT1 over-expressing transgenic mice were generated by use of MICT1 construct produced by insertion of the MICT1 coding sequence with Flag and HA tagged at the C-terminus after EcoRI and SacI digestion of pCAG-loxP-loxP-ZsGreen (Addgene 51269). Transgene insertion was verified by genotyping and sequencing of the MICT1-Flag-HA transgene coding region. Germline transmitted mouse line was crossed with Ucp1-Cre mice (JAX 024670) for MICT1 overexpression in Ucp1+ cells (MICT1-BSOE). MICT1 floxed mice were generated by injection into zygotes with Cas9 nickase mRNA and guide RNA CGCAGAGGCGACGCCATG, along with DNA donor sequence containing two LoxP sites flanking Exon 2 of MICT1. For KO in Ucp1+ cells, MICT1 floxed mice were crossed with Ucp1-Cre mice (MICT1-BSKO). MICT1-BSKO mice were compared to control, fl/fl littermates for all experiments. Primers for genotyping are listed in Reagents and Tools Table. Mice were housed in a 12:12 light-dark cycle and chow (Harlan Teklad LM-485) or HFD (45% of calories from fat, 35% of calories from carbohydrates, and 20% of calories from protein; Dyets), ad libitum, after weaning. No blinding was done for animal experiments. Mice were randomized when allocating animals to drug treatments.

## Cell culture

HEK293FT cells were from UC Berkeley Cell Culture Facility. The immortalized BAT cell line was from Dr. Shingo Kajimura, Harvard University. Human adipose-derived mesenchymal stem cells (PCS-500-011) and MesenCult™ MSC Basal Medium were purchased from ATCC. BAT cells and HEK 293FT cells were maintained in DMEM containing 10% FBS and 1% penicillin/

streptomycin prior to differentiation and transfection. Brown adipocyte differentiation was induced by treatment of confluent cells with DMEM containing 10% FBS, 850 nM insulin, 0.5 mM isobutyl-methylxanthine, 1 µM dexamethasone, 1 nM T3, 125 nM indomethacin and 1 µM rosiglitazone. After 48 h of induction, cells were switched to a maintenance medium containing 10% FBS, 850 nM insulin, 1 nM T3, and 1 µM rosiglitazone. BAT cells were infected with MICT1 adenovirus or lentivirus as indicated on Day 4 of differentiation. Medium was replaced by maintenance medium one day after the infection. Cells were incubated with 10 µM forskolin or 1 µM CL-316,243. MICT1 KO pools were generated by transducing BAT cells that express Cas9 cells with lentivirus containing sgRNA. Lentivirus construct was produced by subcloning annealed guide RNA to U6-stuffer-longtracer-GFP construct. BAT-Cas9 cells were infected with lentivirus containing different gRNAs and were selected for GFP by FACS for MICT1 KO pools. gRNA are; gRNA1-F: TTGGCAGCAACAACAACCACGACG, R: AAACCGTCGTGGTTGTTGTTGCTG gRNA2-F: TTGGCCG TCCTGAATGACAAGCAC, R: AAACGTGCTTGTCATTCAG GACGG.

## Mouse treatment, body temperature measurement, and indirect calorimetry

For the systemic administration, AAV were diluted in 200 µL saline and injected into the lateral tail vein. Body weight was recorded weekly, and mice were subjected to cold exposure for EchoMRI, GTT, and ITT. Core body temperature was determined using a Physitemp BAT-12 probe at 4 °C. Oxygen consumption was measured using the Comprehensive Laboratory Animal Monitoring System (CLAMS). Data were normalized to lean body mass determined by EchoMRI. Mice were individually caged and maintained under a 12 h light/12 h dark cycle. Food consumption and locomotor activity were tracked. For PP2B inhibitor study, 50 µM of INCA-6 (MedChemExpress HY-108544) was directly administered into BAT of HFD fed mice via subcutaneous injection, and experiments were carried out after 24 h.

## Plasmid constructs

pcDNA3.1 +/C-(K)DYK MICT1 plasmid was generated by GenScript. Human MICT1 plasmid was purchased from Origene (RC21684). We subcloned HA-tagged 76 aa MICT1 construct by PCR using: Forward primer (F): ATGGGGCTGAAGCCCTCCT GTCTGAAAGGCTTTAAG Reverse primer (R): CAGACAG-GAGGGCTTCAGCCCCATGGTGGCGGATCCGAGCTC.

To generate various MICT1 mutant constructs, the following primers were used for subcloning: G2A-MICT1 mutation—F: CCACCATGGCCCTGAAGCCCTCCTGTCTGAAAGGC, R: GGC TT CAGGG CCATGGTGGCGGATCCGAGCTCGGTACC; LAV A-MICT1 mutation—F: GACAAGC ACGCCAGCGCGCCCAACA TTATCATCACGCCCCCAA CC, R: GATAATGTTG GGC GCGC TGGCGTGCTTGTCATTCAGGACGGGGGCC; PNAAAT-MICT 1 mutation—F: GAGCGTGCCC AACGCCGCTGCCACGCCCCC AACCCCGACGGGCATG, R: GTTGGGGG CGTGGCAGCGG C GTTGGGCACGCTCAGGTGCTTGTCATTC. Stop Condon mutation—F: ATGGGGCTGAAGC CCTCCTGACTGAAAGGCTTTA AGATGTG, R: CACATCTTAAAGCCTTTCAGTCAGGAGGGC TTCAGCCCCAT.

## Adenovirus, AAV, and lentivirus generation

Ad-MICT1 adenovirus was generated from Vector Biolabs. For knockdown experiments, a lentivirus construct, TRC2-pLKO.5-puro vector containing the shRNA targeting MICT1 was purchased from Sigma (targeting sequence: TCCTGAATGACAAG-CACCTGA). Lentivirus was packaged in HEK 293 FT cells using MISSION® Lentiviral Packaging Mix (Sigma, SHP001), and the viral media was harvested after 48 h for further experiments. AAV constructs were generated by replacing the promoter and gene cassette of AAV-FLIM-AKAR (Addgene, 105903) with 211 bp mini/Ucp1 enhancer (Kozak et al, 1994) and MICT1 or eGFP CDS by MluI and EcoRI digestion. The constructs were then co-transfected into AAVPro293T cells (Takara, 632273) with pHelper and pAAV2/8 (Addgene, 112864). AAV particle was purified by using Takara Purification kit (Takara, 6675).

## PKA activity measurement and AKAR imaging

For biochemical PKA assays, BAT cells pretreated with H89 at 10 μM for 2 h were treated with forskolin or CL 316, 243 for 15 min and assayed by using PKA Colorimetric Activity Kit (Thermo Fisher). For live cell PKA imaging, PKA activity was measured by using excitation-ratiometric PKA activity reporter, ExRai-AKAR2 (Zhang et al, 2022). Upon PKA phosphorylation in live cells, the peak excitation wavelength of cpEGFP shifts from 405 nm to 480 nm. Excitation scans were collected at 520 nm emission for ExRai-AKAR2. To express the AKAR reporter in the differentiated BAT cells, lentiviral construct was generated by subcloning ExRai-AKAR2 (Addgene, 161753) into pCDH lentiviral vector and packaged in HEK293 cells. Differentiated BAT cells were cultured and transduced with ExRai-AKAR2 lentivirus on Day 4 of differentiation. Cells were washed twice before imaging in dark at 37 °C with 5% $CO_2$ on a Zeiss LSM 710 Laser Scanning Confocal Microscope. Dual excitation-ratio imaging was performed by 520 nm emission and 405 nm or 480 nm excitation. Live time-course images were acquired every 15 s. After the baseline reading for 2 min, forskolin at 50 uM was added to measure PKA activity in the stimulated condition. Live cell images were acquired for 10 min.

## RNA-Seq

MICT1 KO cells and Scr BAT cells were treated with forskolin for 6 h. RNA were prepared using RNeasy kit (Qiagen, 74004). Strand-specific libraries were generated from 500 ng RNA using the TruSeq Stranded Total RNA Library Prep Kit (Illumina). cDNA libraries were pair-end sequenced on an Illumina HiSeq 4000. Using Partek software, reads were aligned to the mouse genome (NCBI38/mm10). A gene was included in the analysis if met the following criteria: the maximum RPKM reaching four unit at any time point and the gene length being >100 bp, and level difference of at least +/−1.8-fold with significance ($p < 0.05$).

## Phosphoproteomics

MICT1 KO and Scr BAT cells differentiated and treated with forskolin were lysed in SDS buffer. Each sample was digested according to S-Trap Micro kit (ProtiFi) procedure. Digested peptides were enriched with the Fe-NTA kit (Thermo Fisher,

A32992), labeled with tandem mass tag (TMT) for multiplexed quantitative proteomics and cleaned by using C18 StageTips (Pierce, 87782). Phosphopeptides were analyzed by LC-MS/MS. Protein and peptide identification and TMT based quantification was performed with Proteome Discoverer 2.5 (Thermo Fisher).

## Oxygen consumption rate assay

BAT cells were differentiated in 12-well plates, trypsinized, and reseeded in XFe24 plates at 50 K cells per well on Day 5 and assayed on Day 6 of differentiation. The cells were washed 3 times and maintained in XF-DMEM (Agilent) supplemented with 1 mM sodium pyruvate and 17.5 mM glucose. Oxygen consumption was blocked by 1 μM oligomycin. Maximal respiratory capacity was assayed by the addition of 1 μM FCCP. Uncoupled respiration was calculated as OCR under oligomycin treatment minus OCR under antimycin A/rotenone. BAT tissue was harvested from mice and 9 mg of tissue was excised in one single cut and seeded into the Seahorse Islet Capture microplate wells. The plate was incubated for 1 h at 37 °C without $CO_2$ prior to analysis on the XFe24 Analyzer.

## RT-qPCR analysis and immunoblotting

Reverse transcription was performed with 500 ng of total RNA using SuperScript IV (Invitrogen, 18090010). qPCR was performed in duplicates with BioRad CFX Maestro Software to quantify the relative mRNA levels. Statistical analysis of the qPCR was obtained using the ΔΔCt ($2^{-ddCT}$) method with Eef1a1 as the control. qPCR primer sets are listed in Reagents and Tools Table. For immunoblot (IB) analysis, total cell lysates in RIPA buffer and the plasma membrane fraction isolated using the plasma membrane protein isolation kit (Abcam, ab65400) were separated by SDS-PAGE, transferred to nitrocellulose membrane, and each membrane was cut to 2–3 sections to probe with various antibodies listed in Reagents and Tools Table.

## Co-immunoprecipitation

293FT cells were transfected using Lipofectamine 2000 to express Myc-PP2B and various FLAG-tagged MICT1 constructs (WT, mutants or deletion constructs). Cells were lysed in IP buffer containing 20 mM Tris, pH 7.4, 150 mM NaCl, 1 mM EDTA, 10% glycerol, and 1% NP-40 supplemented with proteases inhibitors. Cell lysates were incubated for 2 h at 4 °C with anti-FLAG M2 or anti-Myc magnetic beads. Magnetic beads were washed 3 times and bound proteins were eluted by boiling in Laemmli sample buffer and analyzed by immunoblotting using the indicated antibodies.

## Whole mount staining

1 mm piece of tissue was excised, fixed with 1% PFA, incubated with UCP1 antibody, LipidTox Red Reagent (Thermo Fisher) and DAPI. Tissues were immobilized on a slide with mounting medium and visualized using a confocal microscope.

## ERthermAC

Differentiated BAT cells were stimulated with forskolin for 6 h, then treated with ERthermAC (250 nM) for 30 min prior to FACS analysis.

## Luciferase reporter assay

HEK293FT cells were co-transfected with −1.6 kb MICT1-Luc promoter and CREB1 expression vector or EV, along with renilla reporter plasmid. Cells were lysed 48 h post-transfection and assayed for luciferase activity using the Dual-Glo Luciferase Kit (Promega). For the mutation study, CREB binding site upstream of MICT1 promoter sequences at −1386 bp: catgtcatcagcccc was mutated to cattcacgactaa; at 1246 bp sequence: aggtgatgtaaca was mutated to aggacactcggtg; at −456 bp sequence: aggtgatgtaaca was mutated to aggcattggaaa.

## Glucose uptake assay

One day prior to the experiment, differentiated BAT cells were trypsinized and reseeded in an opaque 96-well plate at 20 K cells per well in maintenance medium without FBS. On the day of the experiment, cells were treated with 10 μM forskolin for 6 h before being assayed for glucose uptake rate using Glucose Uptake-Glo Assay.

## MitoTracker

For live-cell microscopy, cells were grown in 35 mm glass-bottom imaging dishes (Cellvis). Cells were treated with 1 μM CL-316,243 24 h prior to imaging. To image mitochondria, cells were incubated with 100 nM MitoTracker Green and 1 μg/mL Hoechst for 30 min and washed 3X with PBS, and imaged in fresh medium.

## Quantification and statistical analysis

Statistical comparisons were made using a two-tailed unpaired *t*-test using GraphPad Prism 8 (GraphPad). For genome-wide analyses, Partek Genomics Suite (Partek) was used to produce GSA for RNA-Seq differential expression comparison and Gene set enrichment tool for Gene Ontology to identify the significantly affected pathways. Data is expressed as means ± standard errors of the means (SEM) of indicated biological replicates. The statistical differences in mean values were assessed by Student's *t* test. All experiments were performed at least twice and representative data are shown. All replicates are biological replicates.

### *Graphics*

Graphics for the synopsis image were created with BioRender.com.

## Data availability

Resources and unique reagents generated in this study should be requested to the Lead contact, Hei Sook Sul (hsul@berkeley.edu), with a Materials Transfer Agreement. RNA sequencing data that support the findings of this study were deposited in GEO, accession number: GSE235073 (https://www.ncbi.nlm.nih.gov/geo/query/acc.cgi?acc=GSE235073, reviewer token: wlgduuqypvwvdyp). All source data are available on Biostudies, accession: S-BSST1828 (https://www.ebi.ac.uk/biostudies/studies/S-BSST1828?key=8f030cd7-c41e-4130-a03b-f520fb566f16).

The source data of this paper are collected in the following database record: biostudies:S-SCDT-10_1038-S44318-025-00444-x.

## Peer review information

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

## Acknowledgements

The work is supported partly by NIH grant DK134757 to HSS.

## Author contributions

**Jennie Dinh**: Conceptualization; Data curation; Formal analysis; Validation; Investigation; Visualization; Methodology; Writing—original draft; Writing—review and editing; **Danielle Yi**: Conceptualization; Data curation; Formal analysis; Validation; Investigation; Methodology; Writing—original draft; Writing—review and editing; **Frances Lin**: Data curation; Validation; **Pengya Xue**: Data curation; **Nicholas D Holloway**: Data curation; **Ying Xie**: Data curation; **Nnejiuwa U Ibe**: Data curation; **Hai P Nguyen**: Data curation; **Jose A Viscarra**: Data curation; **Yuhui Wang**: Data curation; **Hei Sook Sul**: Conceptualization; Resources; Supervision; Funding acquisition; Methodology; Writing—original draft; Writing—review and editing.

In addition to the CRediT author contributions listed above, the contributions in detail are: JD performed CL-316,243 studies, performed cAMP and PKA assays and immunoblotting for phosphorylation, assisted in animal studies and AKAR imaging, analyzed the data and wrote the manuscript. DY designed and performed most of the experiments, analyzed the data, and wrote the manuscript. FL performed immunoblotting for phosphorylation, immunofluorescence and mitochondria studies, and assisted AKAR imaging. PX generated plasmid constructs for mouse generation and luciferase constructs. NDH generated luciferase constructs, performed luciferase assay, and immunoblotting for phosphorylation. YX subcloned and generated plasmids for in vitro experiments. NUI assisted in animal cold exposure experiments. HPN assisted in generating floxed mice. Unfortunately, while this paper was under review, this author passed away. The email included is of his department chair. JAV made plasmid constructs for mouse generation. YW guided and assisted in PKA activity assays. HSS designed the project, guided experiments and wrote the manuscript. Source data underlying figure panels in this paper may have individual authorship assigned. Where available, figure panel/source data authorship is listed in the following database record: biostudies:S-SCDT-10_1038-S44318-025-00444-x.

## Disclosure and competing interests statement

The authors declare no competing interests.

# Expanded View Figures

**A**

MICT1

Relative intensity (MICT1/GAPDH)

BAT, iWAT, pWAT, liver, kidney, heart

**B**

Pgc1α

Relative mRNA levels

Day of BAT cell differentiation

Prdm16

Fabp4

**C**

Human cultured thermogenic adipocytes

mRNA abundance (TPM)

p=0.001 p=0.001

p=0.001 p=0.001

p=0.001 p=0.021

MICT1    PGC1α    DIO2

· undifferentiated
· White adipocytes
· Beige adipocytes

**D**

MICT1

Relative intensity (MICT1/GADPH)

p<0.0001

30°C    4°C

**E**

-1.6kb MICT1 promoter region with Creb1 binding motif

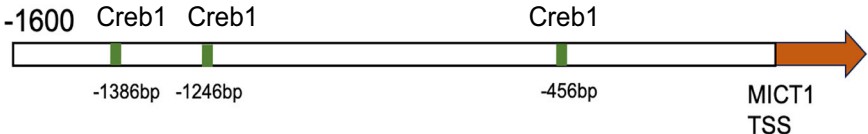

-1600    Creb1    Creb1              Creb1

-1386bp  -1246bp            -456bp    MICT1
                                      TSS

**Figure EV1.  MICT1 is a microprotein highly enriched in brown adipocytes.**

(A) MICT1 protein quantification for various mouse tissues. (B) RT-qPCR for during the course of BAT cell differentiation. (C) Expression of MICT1 in human cultured thermogenic adipocytes, from publicly available RNA-seq data ($n = 50$, White adipocytes MICT1: $P = 0.001$, Beige adipocytes MICT1: $P = 0.001$, White adipocytes PGC1α: $P = 0.001$, Beige adipocytes PGC1α: $P = 0.001$, White adipocytes DIO2: $P = 0.001$). (D) MICT1 protein quantification for mice housed at either 30 °C or 4 °C ($n = 3$) (E) Schematic of CREB sites in the MICT1 promoter. Data is expressed as means ± standard errors of the means (SEM) of indicated number of biological replicates. The statistical differences in mean values were assessed by Student's *t* test.

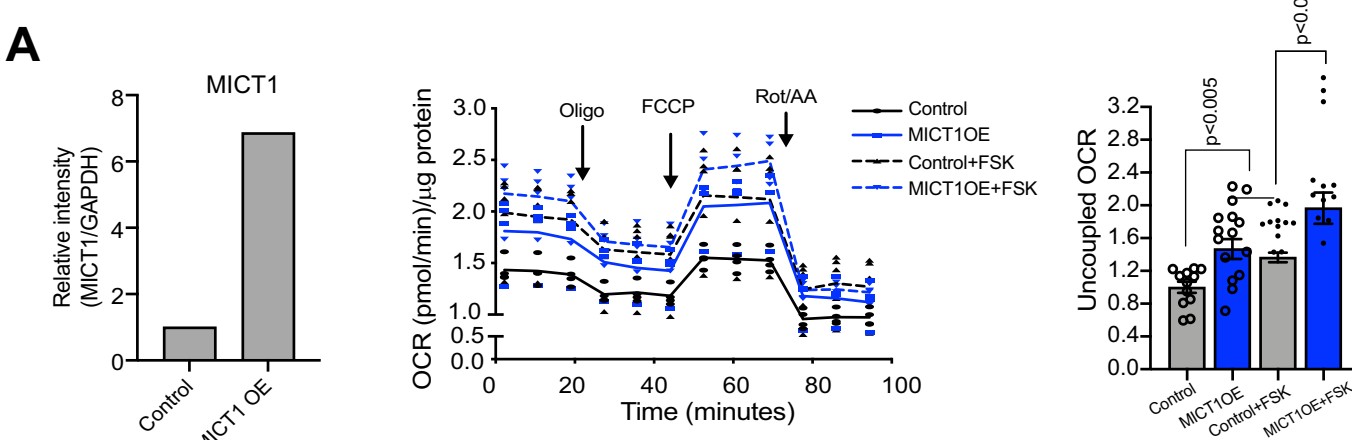

**Figure EV2. MICT1 impact on thermogenesis in cultured brown adipocytes.**

(A) (Left) MICT1 protein quantification in MICT1 OE BAT cells. (Right) OCR measured in MICT1 OE BAT cells that were treated with FSK, and relative uncoupled OCR under oligomycin (0.5 μM) (n = 12, MICT1 OE: P = 0.0041, MICT1 + FSK: P = 0.0009). Data is expressed as means ± standard errors of the means (SEM) of indicated number of biological replicates. The statistical differences in mean values were assessed by Student's t test.

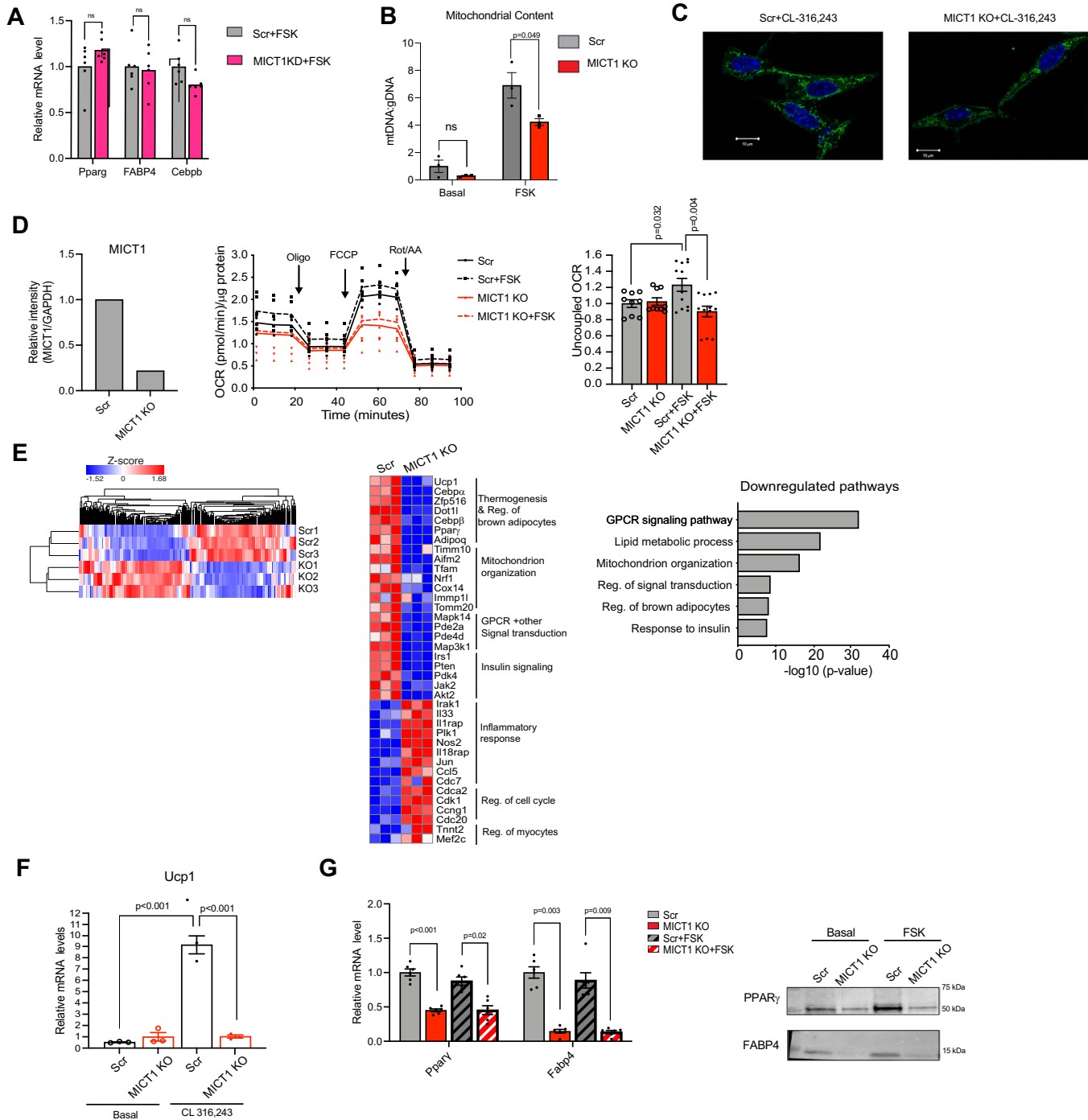

**Figure EV3. MICT1 ablation suppresses thermogenesis in cultured brown adipocytes.**

(A) RT-qPCR in MICT1 KD cells ($n = 6$). (B) Ratio of mtDNA:gDNA of Scr and MICT1 KO cells in the basal and FSK stimulated condition ($n = 3$, MICT1 KO + FSK: $P = 0.0495$). (C) (Left) MICT1 protein quantification in MICT1 KO pool cells. (Right) OCR measured in MICT1 KO pools that were treated with FSK, and relative uncoupled OCR under oligomycin (0.5 μM). (D) (Left) Hierarchical clustering of RNA-seq using differentiated MICT1-KO pools. (Middle) Heatmap showing changes in gene expression in the Scr and MICT1-KO pools. (Right) Representative top GO terms of downregulated genes identified by differential expression analysis. (E) (Left) RT-qPCR for indicated genes in MICT1 KO pools in the basal condition. (Right) RT-qPCR for Ucp1 in MICT1 KO-pools in the basal and CL-316,243 treated conditions. (F) RT-qPCR for MICT1 in Scr and MICT1 KO cells in the basal and CL-316,243-stimulated conditions ($n = 3$, Scr+FSK: $P < 0.0001$, MICT1 KO + FSK: $P < 0.0001$). (G) RT-qPCR ($n = 6$, MICT1 KO Ppary: $P < 0.0001$) and IB for Ppary and Fabp4 in Scr and MICT1 KO cells in the basal and FSK-stimulated conditions. Data is expressed as means ± standard errors of the means (SEM) of indicated number of biological replicates. The statistical differences in mean values were assessed by Student's $t$ test.

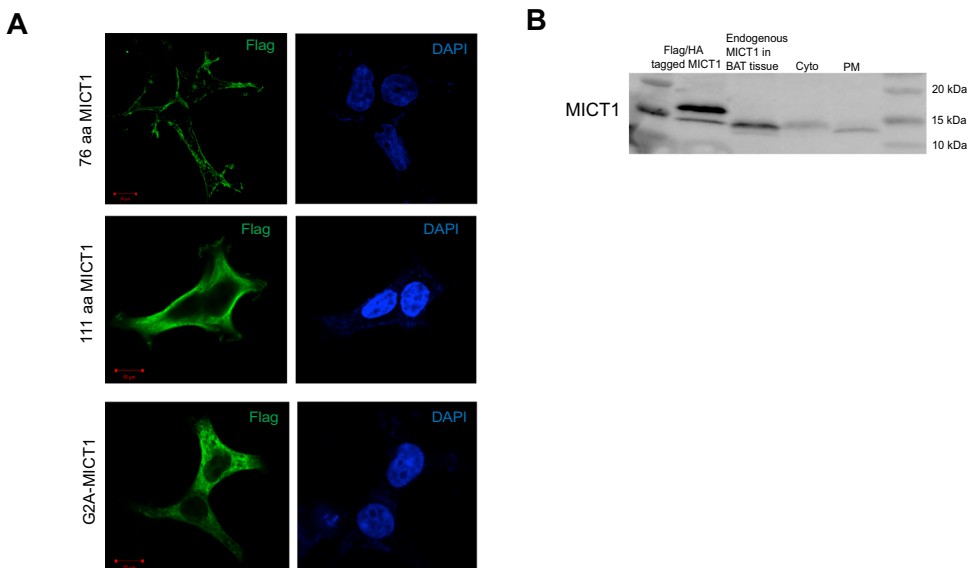

**Figure EV4.  Plasma membrane localization of MICT1 and its interaction with PP2B.**

(**A**) IF images of HEK293FT cells overexpressing 76 aa MICT1 (scale bar: 10 μm), 111 aa MICT1 (scale bar: 10 μm), or G2A-MICT1 (scale bar: 10 μm). (**B**) IB for MICT1 in lysates from HEK293FT cells overexpressing Flag/HA tagged MICT1 (first lane) and endogenous MICT1 in mouse BAT (second lane). MICT1 in the cytosol and plasma membrane of mouse BAT (third and fourth lane).

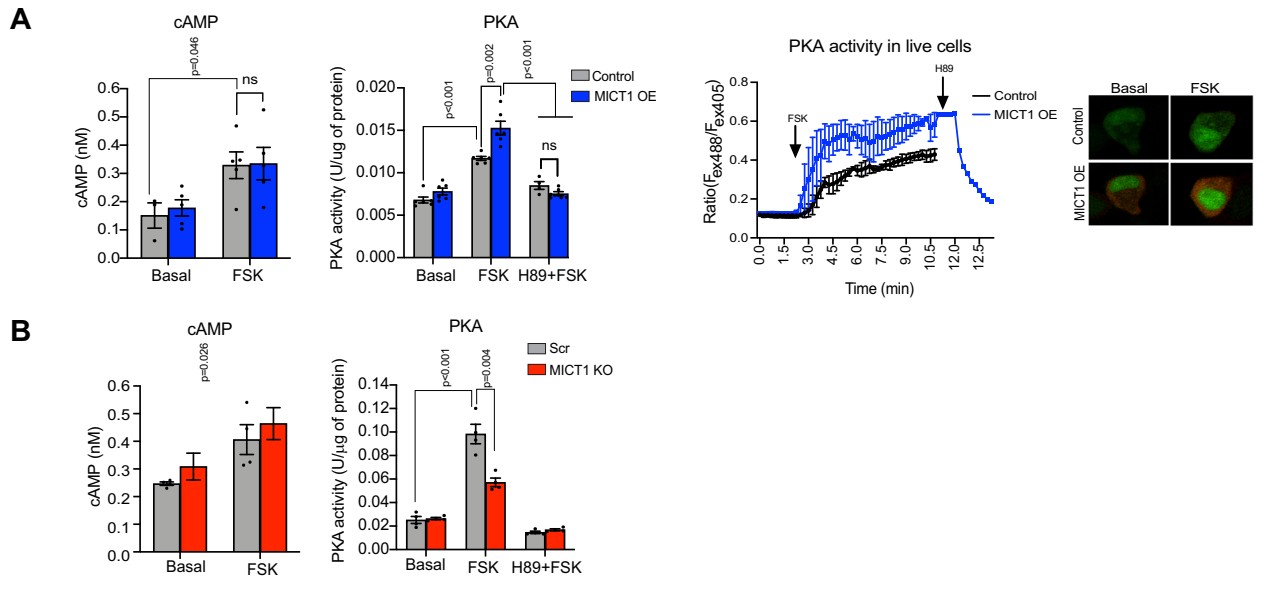

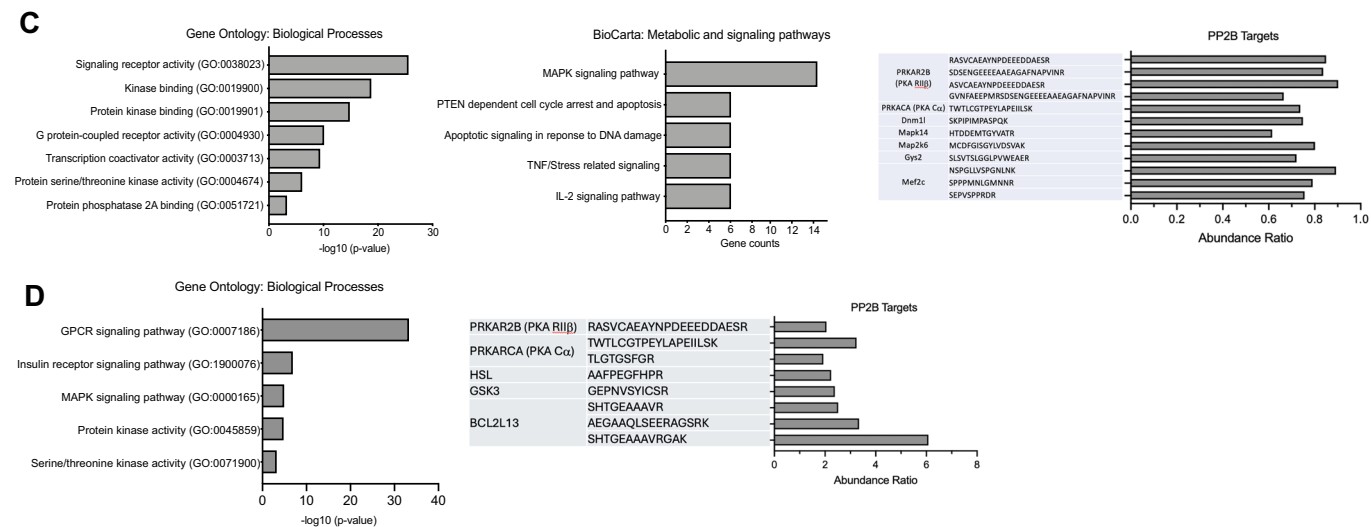

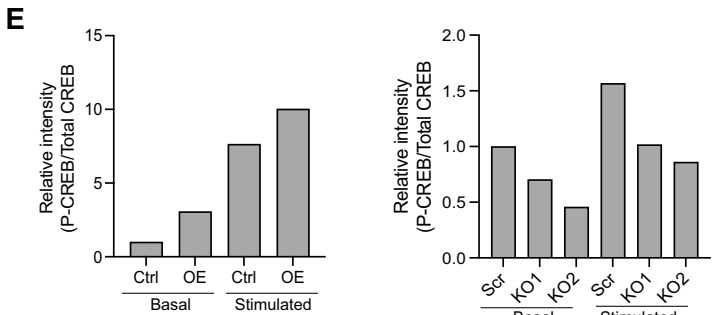

**Figure EV5.   MICT1 potentiates PKA activity.**

(**A**) (Left) Control and MICT1OE in differentiated BAT cells. cAMP levels and PKA activity were measured in the basal condition or after FSK treatment. Cell permeable H89 was used to verify PKA activity ($n = 4$, Control+FSK: $P = 0.0001$, MICT1 OE + H89 + FSK: $P = 0.0001$). (Right) AKAR images of MICT1 OE cells that were treated with FSK. (**B**) Differentiated Scr or MICT1 KO-pools were used to measure cAMP levels and PKA activity in the basal condition or after FSK treatment ($n = 4$, Scr+FSK: $P < 0.0001$). (**C**) Phosphoproteomics analysis of MICT1-CRISPR KO brown adipocytes that were treated with FSK. (Left) Gene Ontology and pathway analysis by BioCarta indicate that PKA pathway is the top signaling pathway affected by MICT1 ablation. (Right) Known PP2B targets with significantly decreased phosphorylation abundance ratio and annotated sequences. (**D**) Phosphoproteomics analysis of MICT1 OE brown adipocytes that were treated with CL-316,243. (Left) Gene Ontology GPCR signaling pathway is the top signaling pathway affected by MICT1 overexpression. (Right) Known PP2B targets with significantly increased phosphorylation abundance ratio and annotated sequences. (**E**) P-CREB protein quantification for MICT1 OE (left) and MICT1-KO pools (right). Data is expressed as means ± standard errors of the means (SEM) of indicated number of biological replicates. The statistical differences in mean values were assessed by Student's $t$ test.

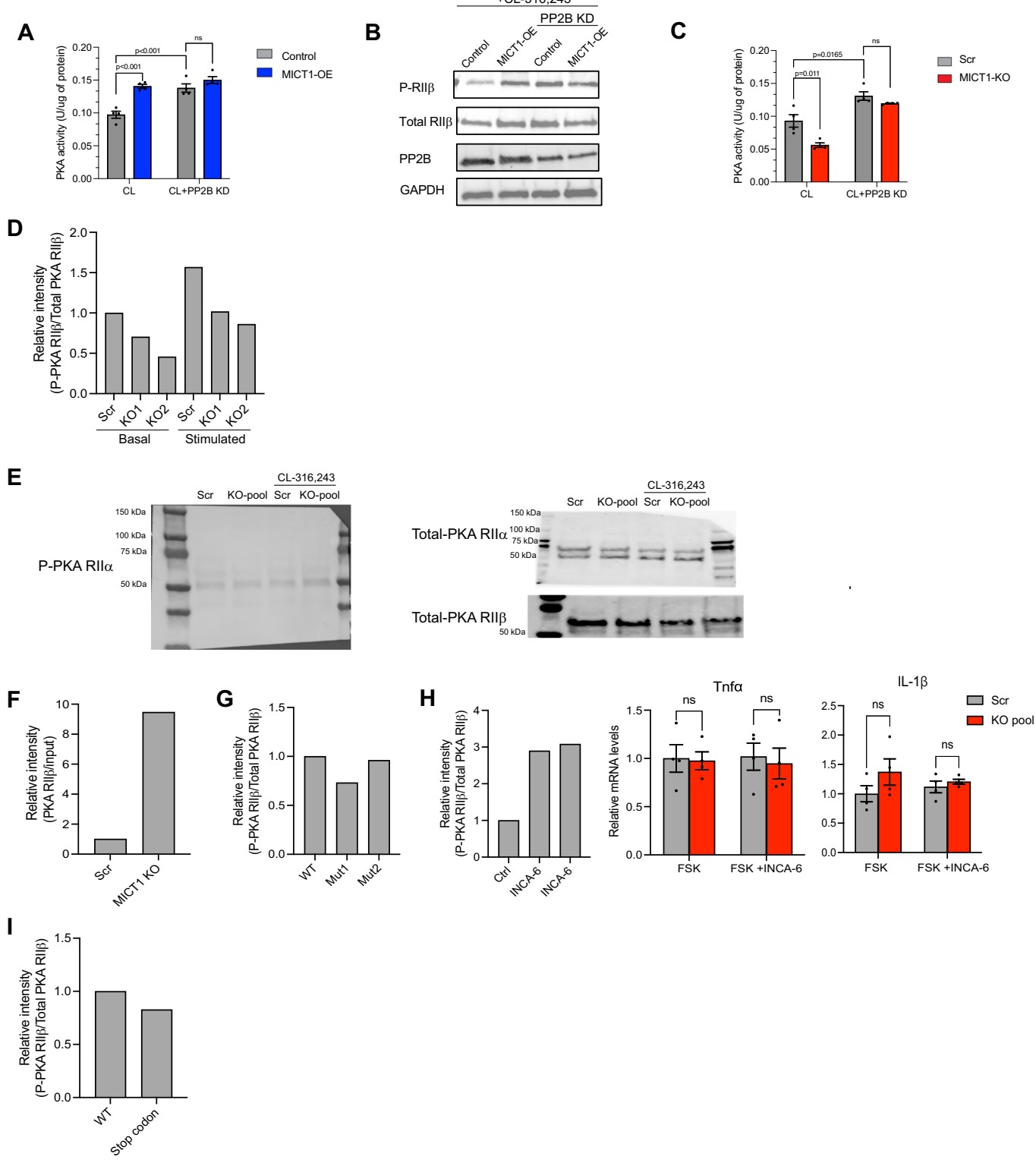

**Figure EV6.  Plasma membrane MICT1-PP2B interaction controls RIIβ dephosphorylation for potentiation of PKA activity and thermogenesis.**

(A) PKA activity ($n = 4$, MICT1 OE + CL: $P = 0.0004$, Control+CL + PP2B KD: $P < 0.0004$) and (B) IB of CL-316,243 stimulated control and MICT1 OE brown adipocytes with or without PP2B. (C) PKA activity of CL-316,243 stimulated Scr and MICT1 KO brown adipocytes with or without PP2B KD ($n = 4$). (D) P-PKA RIIβ protein quantification for MICT1 OE (left) and MICT1-KO pools (right). (E) IB of MICT1-CRISPR KO brown adipocytes that were treated with CL-316,243. (F) Total-PKA RIIβ protein quantification for Scr and MICT1-KO pool lysates that were pulled down with PP2B antibody. (G) P-PKA RIIβ protein quantification for MICT1 mutants. (H) (Left) P-PKA RIIβ protein quantification for INCA-6 injected BAT lysates. (Right) RT-qPCR for IL-1β and Tnfα in FSK-stimulated MICT1-KO BAT cells that were treated with vehicle or INCA-6 (5μM) for 1 h ($n = 4$). (I) P-PKA RIIβ protein quantification for MICT1 with stop codon mutation. Data is expressed as means ± standard errors of the means (SEM) of indicated number of biological replicates. The statistical differences in mean values were assessed by Student's $t$ test.

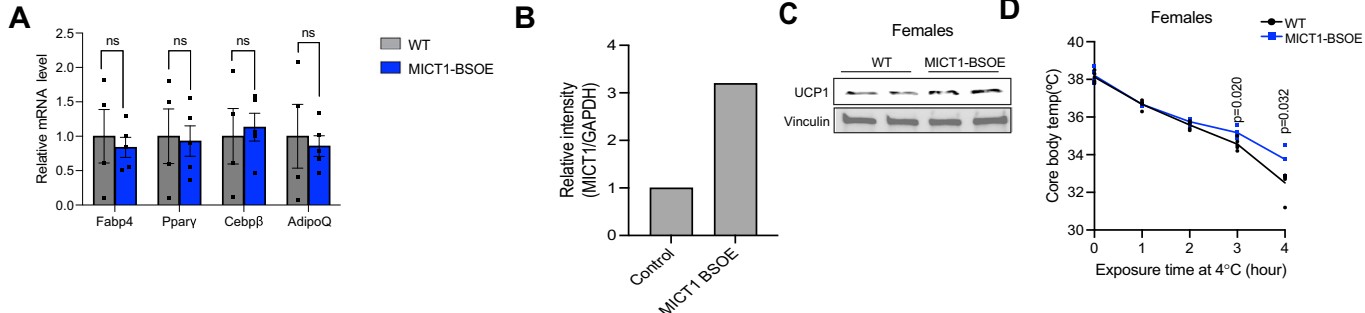

**Figure EV7. MICT1 overexpression in BAT in mice promotes thermogenesis, preventing obesity and insulin resistance.**

(A) RT-qPCR for adipogenic genes in BAT of WT and MICT1-BSOE mice ($n = 4$). (B) MICT1 protein quantification for MICT1-BSOE mice. (C) IB for MICT1 and UCP1 in BAT from MICT1-BSOE and control female mice. (D) Core body temperature measured in 13-wk-old female mice at 4 °C at indicated time points ($n = 4$). Data is expressed as means ± standard errors of the means (SEM) of indicated number of biological replicates. The statistical differences in mean values were assessed by Student's *t* test.

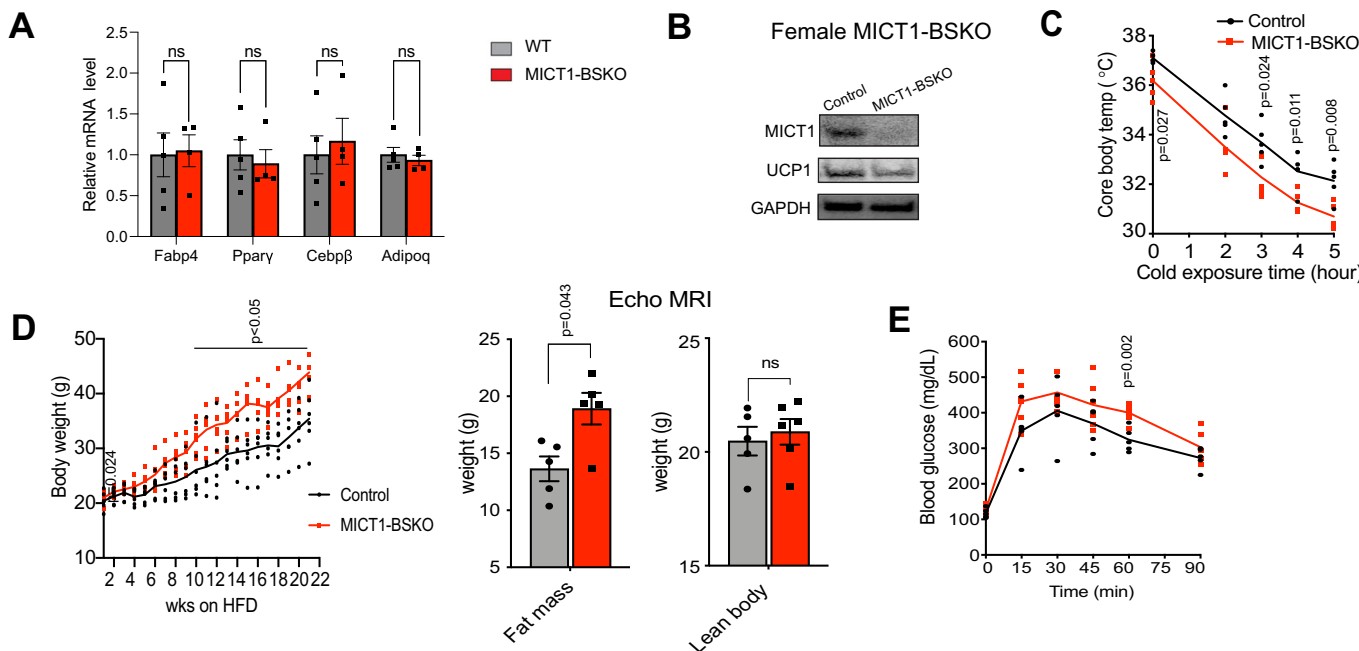

**Figure EV8. MICT1 ablation in BAT in mice reduces thermogenic capacity to gain adiposity.**

(**A**) RT-qPCR for adipogenic genes in BAT of WT and MICT1-BSKO mice (*n* = 4). (**B**) IB for MICT1 and UCP1 in BAT from MICT1-BSKO and control female mice. (**C**) Core body temperature measured in 13-wk-old female mice at 4 °C at indicated time points (*n* = 4 mice per group). (**D**) Body weights and body composition assessed by EchoMRI of control and MICT1-BSKO female mice on HFD (*n* = 6). (**E**) GTT of MICT1-BSKO female mice (*n* = 5). Data is expressed as means ± standard errors of the means (SEM) of indicated number of biological replicates. The statistical differences in mean values were assessed by Student's *t* test.

