## [Peer Review File · The EMBO Journal]

The microprotein C16orf74/MICT1 promotes thermogenesis in brown adipose tissue

Jennie Dinh, Danielle Yi, Frances Lin, Pengya Xue, Nicholas Holloway, Ying Xie, Nnejuwa Ibe, Hai Nguyen, Jose Viscarra, Yuhui Wang, and Hei Sook Sul

Corresponding author(s): Hei Sook Sul (hsul@berkeley.edu)

Review Timeline:

Submission Date:	16th Aug 24
Editorial Decision:	18th Oct 24
Revision Received:	1st Feb 25
Editorial Decision:	6th Mar 25
Revision Received:	17th Mar 25
Editorial Decision:	27th Mar 25
Revision Received:	28th Mar 25
Accepted:	9th Apr 25

Editor: Ieva Gailite

Transaction Report:

Dear Dr. Sul,

Thank you for submitting your manuscript for consideration by the EMBO Journal. I sincerely apologise for the protracted assessment process due to delays in reviewer report submission, We have now received comments from two reviewers, which are included below for your information. Since the third reviewer was not able to submit their report in a timely manner, I am taking the decision based on the reports at hand.

As you will see from the comments, both reviewers appreciate the study and find it per se of interest to the research field. However, they also indicate substantive concerns that would need to be addressed before they can support publication of the manuscript here. Based on overall interest expressed in the reviewer reports, I invite you to revise the manuscript along the lines indicated in the referee comments. Since several of the reviewer comments would require further-reaching experiments, I think it would be helpful to discuss the revision in more detail via email or phone/videoconferencing - please let me know which option you prefer. I should also add that it is The EMBO Journal policy to allow only a single major round of revision and that it is therefore important to resolve the main concerns at this stage.

We generally allow three months as standard revision time, which can be extended up to six months for major revisions. Should you foresee a problem in meeting this deadline, please let us know in advance to discuss an extension.

As a matter of policy, competing manuscripts published during this period will not negatively impact on our assessment of the conceptual advance presented by your study. However, please contact me as soon as possible upon publication of any related work to discuss the appropriate course of action.

When preparing your letter of response to the referees' comments, please bear in mind that this will form part of the Review Process File and will therefore be available online to the community. For more details on our Transparent Editorial Process, please visit our website: <https://www.embopress.org/page/journal/14602075/authorguide#transparentprocess>. Please also see the attached instructions for further guidelines on preparation of the revised manuscript.

Please feel free to contact me if you have any further questions regarding the revision. Thank you for the opportunity to consider your work for publication. I look forward to discussing your revision.

With best regards,

leva

leva Gailite, PhD
Senior Scientific Editor
The EMBO Journal
Meyerhofstrasse 1
D-69117 Heidelberg
Tel: +4962218891309
i.gailite@embojournal.org

- a point-by-point response to the referees' comments, with a detailed description of the changes made (as a word file).

- a word file of the manuscript text.
 - individual production quality figure files (one file per figure)
 - a complete author checklist, which you can download from our author guidelines (<https://www.embopress.org/page/journal/14602075/authorguide>).
 - Expanded View files (replacing Supplementary Information)
- Please see out instructions to authors
<https://www.embopress.org/page/journal/14602075/authorguide#expandedview>
- a Reagents and Tools Table as part of the Methods section, which can be downloaded from our author guidelines (<https://www.embopress.org/page/journal/14602075/authorguide#structuredmethods>)

We realize that it is difficult to revise to a specific deadline. In the interest of protecting the conceptual advance provided by the work, we recommend a revision within 3 months (16th Jan 2025). Please discuss the revision progress ahead of this time with the editor if you require more time to complete the revisions. Use the link below to submit your revision:

Referee #1:

Review of Manuscript # EMBOJ-2024-118785

Thermogenesis is an essential physiological process that maintains body temperature and can further act as an energy dissipative mechanism for metabolic regulation. The brown adipose tissue plays a pivotal role in dissipating energy via non-shivering thermogenesis with UCP1 expression in the mitochondria. Sul and the co-authors report the discovery of a microprotein, MICT1, that can sustain PKA activity by acting as an AKAP for PP2B, preventing the dephosphorylation of the R subunit of PKA and allowing the C subunit of PKA to remain catalytically active. Moreover, MICT1 expression is stimulated by CREB in response to cold stimulus, and its levels correlate with BAT activity and thermogenic gene expression. Using chemical inhibitors, the authors show that the thermogenic ability of MICT1 is mediated through PKA and PP2B, and can be mimicked by INCA-6, which targets the PP2B binding motif of MICT1. Lastly, the authors demonstrate that MICT1 expression and knockout modulates thermogenesis in vivo. By increasing energy expenditure via BAT activation, MICT1 overexpression is sufficient to protect against HFD-induced weight gain, insulin resistance and glucose tolerance. Overall, this is an impressive study going from an uncharacterized gene to an in vivo function with therapeutic implications in metabolic syndrome. That C16ORF74 is a microprotein is tangential to the story, honestly. It's simply another uncharacterized protein that happens to be small. Although the authors provide a detailed mechanism of action to tell a pretty complete story, delineating the function of a novel protein and its regulatory activity over the β 3-AR-PKA axis in thermogenesis, a few points to be addressed prior to make it a strong candidate for EMBO J.

Major points

1. Cyert and colleagues have also discovered this gene to be a PxlIT and LxVP-containing protein is important for tumor growth by regulating the activity of Calcineurin (CN, also known as PP2B). They call it Calcimembrin (<https://doi.org/10.1101/2024.05.12.593783>). I think this should be acknowledged in the text right up front rather than ignoring the fact. The authors should also, in light of this, address why LxVP does not seem to be required for binding in their hands but in the other study it does.
2. Giving multiple names to the same protein causes confusion down the line. I would suggest keeping it as C16ORF74 until HUGO agrees to a nomenclature change in consultation with Cyert and colleagues.
3. Since both studies arrive at PP2B/CN as the binding partner of MICT1/calcimembrin through the use of SLiM identification, I feel that this is a candidate approach and the authors cannot be so sure that MICT1 works solely through PP2B. It would greatly strengthen their conclusions if they could perform an unbiased IP-MS approach to replace the weak Co-IP WB data in Fig 3G to see if PP2B is the main and only target in BAT.
4. Since MICT1 physically binds with PP2B at PNIIT motif, and PP2B has numerous targets for dephosphorylation, have the authors examined the other PP2B downstream targets upon MICT1 OE or ablation? Likewise, the conclusion that MICT1 leads to reduced phosphorylation of PKA targets like HSL is done on a candidate basis, but a better approach is a comprehensive phosphoproteomics experiment to define the targets of both PP2B and PKA that are affected by MICT1 OE or KO. Figure 2e is not so convincing. While Figure 2f does show the proposed trend, it is important to show quantification of repeat experiments.

5. The authors mention in lines 194-5 that MICT1 KO might have affected adipogenesis, and this is also seen in the MICT1 KO animal model where *Ucp1* expression is much lower (Figure 7B). How would that affect PKA activity, or for that matter PKA levels, independently of MICT1 modulation of PKA activity? For instance, they notice that GPCR signaling pathways are downregulated (Figure S2C). That would also result in lower PKA activity following CL /AR stimulation. Their data are also consistent with this scenario. How can they rule this out? Or, does MICT1 affect PKA expression at the protein/transcript level which could also indirectly alter PKA activity?
6. The discrepancy between MICT1 KD and KO adipocyte on thermogenesis and expression of adipogenic genes is puzzling. How do you explain this discrepancy if they do not think MICT1 KO affects differentiation? Could it be residual protein? I could not find a Western blot of MICT1 in the KD BAT cells to compare with that of the KO.
7. The authors conclude that MICT1 potentiates PKA activity by reducing the dephosphorylation of RIIb. Can they show that there as a result, there is less recapturing of the C subunit? This is optional but would really lend credibility to their model.

Minor points

1. Fig 2b, CL treatment increases OCR by around 2 fold. Whereas in Fig 2f and 2i, the OCR increase is minimal (20-40%). Can you address this discrepancy in OCR change with CL treatment?
2. Fig 2g, what is the protein expression level of MICT1 after shRNA KD?
3. y-axis of Fig 6c, d, e does not start from 0. This may be misleading as it makes the differences appear larger visually.
4. Fig 6g, smaller lipid droplets were observed in MICT1-BSOE mice upon HFD. More explanation is needed to address the significance of having smaller lipid droplets and its related beneficial effects.
5. TFAM and PGC1a are downstream of PKA and are important genes for mitochondrial biogenesis. What is the mitochondrial content level in MICT1 OE or KO cells? Any morphological changes to the mitochondria? What is the overall mitochondrial condition in MICT1 OE or KO cells?
6. Western blot figures throughout the manuscript are quite messy and non-aligned, please adjust. There is no need to show the image of the ladder if it causes the panels to be misaligned, just annotate it by the side of the boxed gel image would be sufficient (my personal opinion).
7. Some parts of the text had repeated references to the same data i.e. duplicated sentences. Please revise to avoid repetition.

Examples:

Lines 263-273

H89, the PKA inhibitor, abolished the increase in PKA activity (Fig.S4a, right), 264 verifying PKA activity that we measured. H89 significantly decreased PKA activity, indicating that observed signals indeed displayed PKA activity in live cells.

Lines 180-194

MICT1 knockout cells treated with forskolin had significantly lower expression of thermogenic and mitochondrial biogenesis markers, including *Ucp1*, *Pgc1*, *Tfam*, *Nrf1* and *Cox1* at both mRNA and protein levels (Fig.2j).

We also verified the RNA-seq data by RT-qPCR. MICT1 knockout cells stimulated with forskolin had significantly lower thermogenic and mitochondrial biogenesis markers, while thermogenic genes were unaffected at the basal condition (Fig. 2j). These seem to refer to the same data but sound like different experiments.

Referee #2:

Summary

The study by Yi et al. explores the role of the microprotein MICT1 in brown adipose tissue (BAT) thermogenesis. In this regard, the authors show that the short open reading frame c16orf74, encoding MICT1, was highly enriched in mouse BAT and human beige adipocytes as compared with other tissues. MICT1 expression increased upon cold exposure and beta3-adrenergic stimulation in thermogenic adipocytes. Overexpression of MICT1 in cultured brown adipocytes induced heat production, triggered oxygen consumption rates, and enhanced thermogenic gene expression. Genetic ablation of MICT1 in brown adipocytes reduced heat production, mitochondrial oxygen consumption, and down-regulated transcriptional pathways involved in lipid metabolism, oxidative phosphorylation, and thermogenesis. MICT1 was found to localize at the plasma membrane of Flag-MICT1 transfected HEK cells and murine BAT. In co-immunoprecipitation assays, MICT1 interacted with PP2B (calcineurin) in HEK cells and enhanced intracellular protein kinase A (PKA) signaling including phosphorylation of downstream targets CREB and HSL. MICT1 deficiency led to increased interaction between PP2B and the regulatory subunit RII of PKA, promoting dephosphorylation of RII and PKA inhibition. BAT-specific overexpression of MICT1 in mice enhanced whole-body thermogenesis and protected animals from diet-induced obesity and insulin resistance. AAV-mediated, BAT-selective overexpression of MICT1 in obese ob/ob mice lowered body weight and improved glucose homeostasis. Conversely, BAT-specific MICT1 gene KO impaired systemic energy expenditure, correlating with elevated body weight and impaired glucose homeostasis. The authors conclude that MICT1 may have therapeutic potential in obesity and insulin resistance.

General comments

The mechanisms of BAT thermogenesis remain an active field of research. While the therapeutic use of pro-thermogenic drivers has not been achieved yet, the identification of functionally relevant molecular checkpoints in BAT function is of high interest for the biomedical community. In this respect, the current manuscript by Yi et al. addresses an interesting mechanistic topic. This

holds particularly true for the fact that microproteins remain largely uncharacterized but may harbor therapeutic potential for diverse disease entities. Overall, this study employs state-of-the-art technology, and the conclusions are supported by the experimental data, highlighted by various gain- and loss-of-function models in both animals and cells. The manuscript is concise, well-written and structured. However, a few major concerns still require additional attention by the authors: a) The rationale for specifically studying MICT1 in BAT remains unclear. Neither the introduction nor the discussion provides a clear explanation and a balanced reflection of the available literature to date. C16orf74 has been heavily implicated in pancreatic cancer development, it has been shown to interact with calcineurin before, and even targeting peptides have been explored in the cancer context (e.g. Bradburn, *BioRxiv* 2024; Kushibiki, *Mol Cancer Therap* 2020; Sato, *Oncotarget* 2017). The authors should provide a clear rationale for the reader why and how MICT1 was selected as the protein of interest and reflect upon previous knowledge. An in-depth discussion of its calcineurin interaction appears particularly necessary to substantiate the novelty of the current findings. b) The MICT1-PP2B interaction must be confirmed in BAT in vivo to support relevance. c) In the genetic, gain- and loss-of-function mouse models, does MICT1 regulation impact adipogenesis beyond thermogenesis? Detection of relevant markers of adipocyte differentiation/proliferation status may be sufficient to make conclusions here. d) Do MICT1 levels in human BAT/beige fat correlate with markers body weight, glucose homeostasis or insulin sensitivity? The addition of relevant human data will significantly strengthen the case for publication.

Specific comments

Fig. 1: Please validate the induction of MICT1 protein expression upon cold exposure by Western Blot.

Fig.. 2: Why does MICT1 overexpression only mediate functional effects in brown but not white adipocytes? Please discuss.

Fig. 5: Please validate the functional role of PP2B in MICT1-dependent PKA activation by genetic tools, e.g. siRNA-mediated PP2B knockdown.

Referee #1

1. Cyert and colleagues have also discovered this gene to be a PxIxIT and LxVP-containing protein is important for tumor growth by regulating the activity of Calcineurin (CN, also known as PP2B). They call it Calcimembrin (<https://doi.org/10.1101/2024.05.12.593783>). I think this should be acknowledged in the text right up front rather than ignoring the fact. The authors should also, in light of this, address why LxVP does not seem to be required for binding in their hands but in the other study it does.

We thank the reviewer for bringing this preprint to our attention. We were unaware of this preprint, published on 5/12/2024.

We are glad to read that the preprint also agrees with our result that MICT1 (C16orf74) interacts with PP2B through the PxIxIT motif (Fig.5h-j).

2. Giving multiple names to the same protein causes confusion down the line. I would suggest keeping it as C16ORF74 until HUGO agrees to a nomenclature change in consultation with Cyert and colleagues.

We agree with the reviewer and indicate both names and also mention calcimembrin in this revision.

3. Since both studies arrive at PP2B/CN as the binding partner of MICT1/calcimembrin through the use of SLiM identification, I feel that this is a candidate approach and the authors cannot be so sure that MICT1 works solely through PP2B. It would greatly strengthen their conclusions if they could perform an unbiased IP-MS approach to replace the weak Co-IP WB data in Fig 3G to see if PP2B is the main and only target in BAT.

We appreciate and agree with the reviewer's point. While there may be other MICT1 interacting proteins and would be interesting, the focus of our current study is on the MICT1-PP2B interaction for thermogenesis in brown adipocytes. If there are indeed other MICT1 interacting proteins, studying these proteins is beyond the scope of our study of MICT1 function in brown adipose thermogenesis.

4. Since MICT1 physically binds with PP2B at PNIIT motif, and PP2B has numerous targets for dephosphorylation, have the authors examined the other PP2B downstream targets upon MICT1 OE or ablation? Likewise, the conclusion that MICT1 leads to reduced phosphorylation of PKA targets like HSL is done on a candidate basis, but a better approach is a comprehensive phosphoproteomics experiment to define the targets of both PP2B and PKA that are affected by MICT1 OE or KO. Figure 2e is not so convincing. While Figure 2f does show the proposed trend, it is important to show quantification of repeat experiments.

We agree that it would be interesting to address this in an unbiased approach. In our original submission, we included phosphoproteomics comparing control Scr and MICT1 knockout (KO) brown adipocytes (Fig.S4c). According to the reviewer's suggestion, further analysis of this phosphoproteomics data showed other PP2B downstream targets to be significantly lowered, such as DNML1, GYS2, and MEF2C. We presented the protein names in this revision (Fig.S4c). During this revision, we also performed phosphoproteomics comparing control and MICT1 OE brown adipocytes and found higher phosphorylation of PRKAR2B, PRKARCA, and HSL, as well as some other PP2B downstream targets, including GSK3 and BCL2L13. The new results are also included in this revision (Fig.S4d).

Regarding Fig.2e and Fig.2f, the reviewer may have misread the panels. To clarify, Fig.2e was presented as "negative control data" showing that MICT1 did not promote thermogenesis in white adipocytes. Also, Fig.2f shows quantification of uncoupled oxygen consumption.

5. The authors mention in lines 194-5 that MICT1 KO might have affected adipogenesis, and this is also seen in the MICT1 KO animal model where Ucp1 expression is much lower (Figure 7B). How would that affect PKA activity, or for that matter PKA levels, independently of MICT1 modulation of PKA activity? For instance, they notice that GPCR signaling pathways are downregulated (Figure S2C). That would also result in lower PKA activity following CL /AR stimulation. Their data are also consistent with this scenario. How can they rule this out? Or, does MICT1 affect PKA expression at the protein/transcript level which could also indirectly alter PKA activity?

Our focus was on MICT1 function on brown adipocytes. MICT1 is expressed in adipocytes but not in SVF cells that contain preadipocytes and thus MICT1 would not be able to affect differentiation (Fig. 1b). Thus, we performed MICT1 OE and KD after adipocyte differentiation in vitro. Only exception was MICT1 KO cells generated by CRISPR ablation to delete MICT1 gene at the genomic level. We believe that differentiation of MICT1 KO cells in vitro was affected because we used IBMX, which increases cAMP/PKA, according to the well-accepted in vitro adipocyte differentiation protocol. The role of PKA in adipogenesis in vivo, however, is not clear or known. During the revision, we detected greatly lower R2b phosphorylation in MICT1 KO mice, but no changes in expression of adipogenic markers, such as Pparg, Fabp4, C/ebp β , and AdipoQ in BAT of UCP1 promoter driven MICT1-KO mice and the results are now included in this revision (Fig.7g, S6a, S7a).

However, we appreciate the reviewer's concern and now clarify this point of potential impact of MICT1 deletion at the genomic level for in vitro adipocyte differentiation.

We appreciate and agree with the reviewer's concern on GPCR signaling, the RNA-seq was performed using CRISPR KO cells and we do not know how GPCR signaling was downregulated, although there could be feedback regulation.

6. The discrepancy between MICT1 KD and KO adipocyte on thermogenesis and expression of adipogenic genes is puzzling. How do you explain this discrepancy if you do not think MICT1 KO affects differentiation? Could it be residual protein? I could not find a Western blot of MICT1 in the KD BAT cells to compare with that of the KO. As we explained in #5 above, MICT1 CRISRE-KO cell differentiation in vitro was affected. However, we are focusing on the effect of MICT1 on thermogenesis in brown adipocytes, we performed MICT1 KD post-differentiation, and thus differentiation could not be affected from MICT1 KD. According to the reviewer's

concern, we now include in immunoblot showing decrease in MICT1 (76aa), which is the form in mouse closest with the human ortholog, in MICT1 KD brown adipocytes (Fig 2g, right).

7. The authors conclude that MICT1 potentiates PKA activity by reducing the dephosphorylation of RIIb. Can they show that there as a result, there is less recapturing of the C subunit? This is optional but would really lend credibility to their model.

We thank the reviewer that this is an optional experiment, since recapturing C is not an easy endeavor. We have shown by immunoblotting that MICT1 OE increased phosphorylation of RII α , which implies that there is less recapturing of the C subunit.

Minor points

1. Fig 2b, CL treatment increases OCR by around 2 fold. Whereas in Fig 2f and 2i, the OCR increase is minimal (20-40%). Can you address this discrepancy in OCR change with CL treatment?

Thank you for this comment. We have repeated the experiments and indeed CL-316,243 treatment increases OCR approximately 2-fold in Scr cells and the new result are replaced in Fig.2f and 2i.

2. Fig 2g, what is the protein expression level of MICT1 after shRNA KD?

We include in immunoblot showing decrease in mouse MICT1 (76aa) is the closest human MICT1 ortholog, marked with star in the western blot, in MICT1 KD brown adipocytes (Fig 2g, right).

3. y-axis of Fig 6c, d, e does not start from 0. This may be misleading as it makes the differences appear larger visually.

Thank you for this comment. We have now changed the y-axes for Fig 6D and 6E to begin from 0 for clarity. For Fig 6C, we have inserted a graph with the y-axis from 0 in addition to the original graph.

4. Fig 6g, smaller lipid droplets were observed in MICT1-BSOE mice upon HFD. More explanation is needed to address the significance of having smaller lipid droplets and its related beneficial effects.

Thank you for this comment. BAT thermogenesis employs lipolysis and FAs released are used to fuel thermogenesis. Smaller LD size enhances lipolysis as its increased surface/volume providing surface for lipolytic enzymes to bind. We now include this explanation in the revised manuscript.

5. TFAM and PGC1a are downstream of PKA and are important genes for mitochondrial biogenesis. What is the mitochondrial content level in MICT1 OE or KO cells? Any morphological changes to the mitochondria? What is the overall mitochondrial condition in MICT1 OE or KO cells?

We measured mtDNA in Scr or MICT1 KO BAT cells upon forskolin stimulation, MICT1 KO cells have significantly lower mtDNA compared to Scr cells. Similarly, MitoTracker staining showed significantly lower fluorescence in MICT1 KO cells compared to Scr cells upon CL-316,243 stimulation. This result is now included in the manuscript in Fig.S2c.

6. Western blot figures throughout the manuscript are quite messy and non-aligned, please adjust. There is no need to show the image of the ladder if it causes the panels to be misaligned, just annotate it by the side of the boxed gel image would be sufficient (my personal opinion).

According to the suggestion, we have now re-aligned Western blot figures.

7. Some parts of the text had repeated references to the same data i.e. duplicated sentences. Please revise to avoid repetition.

Examples:

Lines 263-273

H89, the PKA inhibitor, abolished the increase in PKA activity (Fig.S4a, right), 264 verifying PKA activity that we measured.

H89 significantly decreased PKA activity, indicating that observed signals indeed displayed PKA activity in live cells.

Lines 180-194

MICT1 knockout cells treated with forskolin had significantly lower expression of thermogenic and mitochondrial biogenesis markers, including Ucp1, Pgc1a, Tfam, Nrf1 and Cox1 at both mRNA and protein levels (Fig.2j).

We also verified the RNA-seq data by RT-qPCR. MICT1 knockout cells stimulated with forskolin had significantly lower thermogenic and mitochondrial biogenesis markers, while thermogenic genes were unaffected at the basal condition (Fig. 2j).

These seem to refer to the same data but sound like different experiments.

Thank you for this comment. Lines 263-273 are not the same experiments. H89 was used in Fig.S4a to confirm that the assay detecting PKA activity in cell extracts, while in Fig.4b H89 was used to verify detecting of PKA activity in live cells. According to the reviewer, we revised these lines and removed the repetition in 180-194.

Referee #2:

a) The rationale for specifically studying MICT1 in BAT remains unclear. Neither the introduction nor the discussion provides a clear explanation and a balanced reflection of the available literature to date. C16orf74 has been heavily implicated in pancreatic cancer development, it has been shown to interact with calcineurin before, and even targeting peptides have been explored in the cancer context (e.g. Bradburn, *BioRxiv* 2024; Kushibiki, *Mol Cancer Therap* 2020; Sato, *Oncotarget* 2017). The authors should provide a clear rationale for the reader why and how MICT1 was selected as the protein of interest and reflect upon previous knowledge. An in-depth discussion of its calcineurin interaction appears particularly necessary to substantiate the novelty of the current findings.

We found C16orf74 (MICT1) to be an unknown microprotein most highly expressed only in brown adipose tissue, compared to all other tissues, a depot known to be associated with improvements in insulin sensitivity. MICT1 has SNPs associated with Type 2 diabetes according to GWAS. Moreover, there has been no report on microprotein function in thermogenesis. As the reviewer mentioned, C16orf74 has been shown to be elevated in pancreatic ductal adenocarcinoma (PDAC) cells (Kushibiki, 2020; Sato, 2017). Additionally, we thank the reviewer for bringing the preprint to our attention. We were unaware of this preprint, published on 5/12/2024. We are glad to read that the preprint also agrees with our result that C16orf74 interacts with PP2B through the PxIxIT motif (Fig.5h-j). According to the reviewer's suggestion, we now discuss this and added these references in the revised manuscript.

b) The MICT1-PP2B interaction must be confirmed in BAT in vivo to support relevance. For this revision, we used Flag beads to immunoprecipitate MICT1 from BAT lysate of transgenic mice overexpressing Flag-tagged MICT1 followed by immunoblotting using PP2B antibody in vivo to show MICT1-

PP2B interaction in mice using BAT (Fig.3h). Unfortunately, we could not perform the reverse as we found that commercially available PP2B antibodies were not IP-grade.

c) In the genetic, gain- and loss-of-function mouse models, does MICT1 regulation impact adipogenesis beyond thermogenesis? Detection of relevant markers of adipocyte differentiation/proliferation status may be sufficient to make conclusions here.

The role of PKA in adipogenesis in vivo, however, is not clear or known. During the revision, we detected no changes in expression of adipogenic markers, such as Pparg, Fabp4, C/ebp β , and AdipoQ in BAT of UCP1 promoter driven MICT1 OE and KO mice and the results are now included in this revision (Fig.7g, S6a,S7a).

d) Do MICT1 levels in human BAT/beige fat correlate with markers body weight, glucose homeostasis or insulin sensitivity? The addition of relevant human data will significantly strengthen the case for publication.

We do not have access to human BAT/beige fat so we could not determine this. However, we differentiated human beige preadipocytes into beige adipocytes capable of thermogenesis to test this. We found that overexpression of MICT1 enhanced glucose uptake rate and increased insulin sensitivity by Phospho-Akt immunoblotting. Conversely, MICT1 knockdown decreased glucose uptake rate in human beige adipocytes.

Specific comments

Fig. 1: Please validate the induction of MICT1 protein expression upon cold exposure by Western Blot. We now include an immunoblot showing induction of MICT1 protein levels upon cold exposure in Fig.1e.

Fig. 2: Why does MICT1 overexpression only mediate functional effects in brown but not white adipocytes? Please discuss.

MICT1 overexpression increases thermogenic activity and thermogenic gene expression in brown adipocytes and beige adipocytes, that have thermogenic potential. White adipocytes do not express UCP1 and thus cannot perform UCP1-dependent thermogenesis. We now clearly discuss this in the revision.

Fig. 5: Please validate the functional role of PP2B in MICT1-dependent PKA activation by genetic tools, e.g. siRNA-mediated PP2B knockdown.

As the reviewer suggested, we performed PP2B knockdown and verified the role of PP2B in MICT1-dependent potentiation of PKA activity. This result is now shown in Fig.S5a-c.

Dear Dr. Sul,

Thank you for submitting a revised version of your manuscript. We have now received input from two of the original reviewers, who find that their main concerns have been addressed satisfactorily.

I will therefore be happy to accept the manuscript for publication after addressing the final minor points outlined by both reviewers.

Additionally, there remain a few editorial points that need addressing before I can extend official acceptance of the manuscript:

1. Figure 2 is currently distributed over two pages. Since all panels would have to fit on a single A4 page to be typeset, I would suggest turning them into two separate figures.
2. CRediT has replaced the traditional author contributions section because it offers a systematic, machine-readable author contributions format that allows for more effective research assessment. Please remove the Authors Contributions from the manuscript and use the free text boxes beneath each contributing author's name in our online submission system to add specific details on the author's contribution. More information is available in our guide to authors.
3. Please update references according to The EMBO Journal style - where there are more than 10 authors on a paper, the first 10 should be listed, followed by 'et al.' Please see further information here:
<https://www.embopress.org/page/journal/14602075/authorguide#referencesformat>
4. To acknowledge the use of BioRender, please add a dedicated section to the Methods along the following format:
"Graphics:

Graphics for the synopsis image were created with BioRender.com."

5. All Materials and Methods need to be described in the main text using our 'Structured Methods' format. According to this format, the Methods section includes a Reagents and Tools Table (listing key reagents, experimental models, software and relevant equipment and including their sources and relevant identifiers) followed by a Methods section describing the methods, ideally using a step-by-step protocol format. The aim is to facilitate adoption of the methodologies across labs.

Please download and fill our Reagents and Tools Table template (.docx), which you can find in our author guidelines:

<https://www.embopress.org/page/journal/14602075/authorguide#structuredmethods>

It appears that the information currently included in Tables S1 and S2 should form a part of the Reagents and Tools table.

6. Individual figure panels Fig 6A and Fig 7K are not mentioned in the manuscript text - please add the corresponding callouts.

7. While screening through the provided Western blot source data, I noted that several figure panels representing this type of data appear vertically compressed in comparison to the original images. Please edit the corresponding figure panels to more accurately represent the original dimensions of the bands, as this can alter the perception of the signal density and strength.

8. Please provide source data for the Western blot data in figure 1e.

9. Our data editors have flagged the following issues in figure legends that need correcting:

- Please note that the expanded view sub-figure 6D is mislabeled as expanded view figure 6C in the manuscript. This needs to be rectified.
- Please provide the exact p values in the legends of figures 1D, E, F, G; 2A, C, E-I; 4A, C; 5A, B, C, D, E; 6A, E; 7A, H, I, K; EV1 C; EV2 A, E, F; EV4 A, B; EV5 A; EV7 D
- Please indicate the statistical test used for data analysis in the legends of figures 1B, D, E, F, G; 2A-D, E-I; 4A, C; 5A, B, C, D, E, F, K, L; 6A-F, H-M; 7A, B, C, D, E, F, H, I, K; EV1 C; EV2 A, B, C, E, F; EV4 A, B, C, D; EV5 A, C, H; EV6 A, D; EV7 A, C, D, E.
- Please provide information on the number and nature of replicates in the legends of figures 1A, B, C, F, G, 2A, B, D, E, G-I; 4A, B, D; 5C, D, E, F, L; 6B, E, F, H, I, L, M; 7B, E, F, I, K; EV1 C, EV2 A, B, C, E, F; EV5 A, C, H; EV6 A, EV7 A, D
- Please define the error bars in the legends of figures 1A-G; 2A-I; 4A-D; 5A-D, E, F, K, L; 6A-F, H, I, J, K, L, M; 7A, B, C, D, E, F, H, I, K; EV1 C; 2 A, B, C, E, F; EV4 A, B; EV5 A, C, H; EV6 A, D; EV7 A, D.
- Please note that scale bar and its definition are missing for figures 3B, D, G; EV3 A.

With best wishes,

leva

We realize that it is difficult to revise to a specific deadline. In the interest of protecting the conceptual advance provided by the work, we recommend a revision within 3 months (4th Jun 2025). Please discuss the revision progress ahead of this time with the editor if you require more time to complete the revisions. Use the link below to submit your revision:

Referee #1:

The authors have addressed most of my scientific concerns, and thank you for clarifications on some of the panels. I would request that the title contains C16ORF74 (MICT) since that is the official gene nomenclature. The field is currently trying to move away from the multi-unofficial naming of microproteins. Discussion of Cyert and colleagues work should also move up to be referenced in the introduction (eg. when summarizing findings of PP2B interaction).

Referee #2:

The authors added a significant amount of new data and improved the manuscripts. In my view, two issues remain to be resolved:

- a) The WB in Figure 1e is clearly not sufficient to demonstrate MICT1 induction in BAT. Please show at least an n=3 and provide a densitometric analysis.
- b) The issue why MICT1 is not functional in WAT remains obscure. It is acknowledged that WAT is not capable of UCP1-mediated thermogenesis. However, all principle components of the MICT1 pathway (PKA) are also present in white adipocytes. Why they differ in their response as compared with brown and beige adipocytes remains a major unresolved question and should be clearly addressed in the discussion.

Referee #1:

The authors have addressed most of my scientific concerns, and thank you for clarifications on some of the panels.

I would request that the title contains C16ORF74 (MICT) since that is the official gene nomenclature. The field is currently trying to move away from the multi-unofficial naming of microproteins. Discussion of Cyert and colleagues work should also move up to be referenced in the introduction (eg. when summarizing findings of PP2B interaction).

As suggested, we have modified the title to “The microprotein C16orf74 (MICT1) promotes thermogenesis” and included of Cyert and colleagues to the introduction as well as in discussion as suggested by the reviewer.

Referee #2:

The authors added a significant amount of new data and improved the manuscripts. In my view, two issues remain to be resolved:

a) The WB in Figure 1e is clearly not sufficient to demonstrate MICT1 induction in BAT. Please show at least an n=3 and provide a densitometric analysis.

According to the reviewer's comment, we have now included immunoblot to show MICT1 induction with n=3 and quantified MICT1 protein levels (Fig.1e, Fig.EV1d).

b) The issue why MICT1 is not functional in WAT remains obscure. It is acknowledged that WAT is not capable of UCP1-mediated thermogenesis. However, all principle components of the MICT1 pathway (PKA) are also present in white adipocytes. Why they differ in their response as compared with brown and beige adipocytes remains a major unresolved question and should be clearly addressed in the discussion.

As the reviewer mentioned, white adipocytes are not capable of UCP1-mediated thermogenesis although PKA pathway is present. While MICT1 potentiates PKA activity, treatment with norepinephrine for 6 hours as in Fig. 2d is not sufficient to induce thermogenesis. Only when white adipocytes in culture are treated with beiging agents T3 and Rosiglitazone, these cells become beige adipocytes capable of thermogenesis and when these beige adipocytes treated with norepinephrine MICT1 overexpression can increase thermogenesis (Fig.8h). This explanation is now included in the result section as suggested by the reviewer.

Dear Hei-Sook,

Thank you for submitting a reformatted version of your manuscript. I have now gone through the revised version, and I am afraid that there remain a few editorial aspects as outlined below that still need to be implemented in the manuscript before its acceptance:

1. In our standard source data check, we have noted unexplained numerical duplications in the source data for several figures. I have attached the corresponding files with the detected duplications labelled in colour. Please take a look and correct if needed. A brief explanation would be very helpful - I appreciate that these duplications can also occur due to specific measurement or calculation methods used.
2. Please update the reference style throughout the manuscript text. They are currently numerical, while our style requires citations by author and year of publication, i.e. Smith & Jones, 2003; Smith et al, 2000. Further information is available here: <https://www.embopress.org/page/journal/14602075/authorguide#referencesformat>
3. The scale bar for figures 4b,d,e,g is poorly visible, while it is not available for EV3c. Please increase the thickness of the scale bar and consider changing the colour to white for better visibility.
4. We would like to propose minor textual edits in the manuscript title, synopsis and abstract. I have also written a short blurb that will accompany the title of your manuscript in our online table of contents. Please take a look at the text below and in the attached file and let me know if any edits or corrections are needed.

Title:

The microprotein C16orf74/MICT1 promotes thermogenesis in brown adipose tissue

Blurb:

The murine MICT1 (microprotein for thermogenesis 1) enhances thermogenesis via PKA activation and protects against adiposity and insulin resistance.

Synopsis:

Microproteins are increasingly found to contribute to various physiological processes. This study identifies MICT1 (microprotein for thermogenesis 1) as a microprotein highly expressed in brown adipose tissue, where it enhances thermogenesis and protects against adiposity.

- MICT1/C16orf74 is specifically and highly expressed in brown adipose tissue and is induced upon cold exposure.
- MICT1 enhances thermogenesis via an interaction with PP2B that disrupts dephosphorylation of PKA-R11b.
- Ucp1⁺-cell-specific ablation of MICT1 in mice suppresses thermogenic capacity to increase adiposity and insulin resistance.
- MICT1 overexpression in UCP1⁺-cells enhances thermogenesis and increases energy expenditure to protect against genetic and diet-induced obesity and insulin resistance.

Please feel free to contact me if have any questions regarding these final points. I look forward to hearing from you.

With best wishes,

leva

leva Gailite, PhD
Senior Scientific Editor
The EMBO Journal
Meyerhofstrasse 1
D-69117 Heidelberg
Tel: +4962218891309
i.gailite@embojournal.org

We realize that it is difficult to revise to a specific deadline. In the interest of protecting the conceptual advance provided by the work, we recommend a revision within 3 months (25th Jun 2025). Please discuss the revision progress ahead of this time with the editor if you require more time to complete the revisions. Use the link below to submit your revision:

All editorial and formatting issues were resolved by the authors.

Dear Hei Sook,

Thank you very much for clarifying the final editorial concerns and for your final textual corrections. I have now gone through the files, and everything looks in order. I am now pleased to inform you that your manuscript has been accepted for publication in the EMBO Journal. Congratulations on a nice study!

As a final minor point, I noticed that in Figures 4c and 4f, the black boxes around the blots have become shifted during editing. You can send me the corrected file via email to make sure that the typesetting process runs smoothly.

If you have any questions, please do not hesitate to contact the Editorial Office. Thank you for your contribution to The EMBO Journal!

With best wishes,

Ieva

Ieva Gailite, PhD
Senior Scientific Editor
The EMBO Journal
Meyerohofstrasse 1
D-69117 Heidelberg
Tel: +4962218891309
i.gailite@embojournal.org
